# Complex evolution in *Aphis gossypii* group (Hemiptera: Aphididae), evidence of primary host shift and hybridization between sympatric species

Yerim Lee[1], Thomas Thieme[2], Hyojoong Kim[1]*

**1** Animal Systematics Laboratory, Department of Biology, Kunsan National University, Gunsan, Republic of Korea, **2** BTL Bio-Test Labor GmbH Sagerheide, RG Phyto-Entomology, Gross Lüsewitz, Germany

\* hkim@kunsan.ac.kr

**Data Availability Statement:** All relevant data are within the manuscript and its Supporting Information files (input files included in ZIP file).

**Funding:** The Korea Environment Industry & Technology Institute (KEITI) through Exotic

## Abstract

Aphids provide a good model system to understand the ecological speciation concept, since the majority of the species are host-specific, and they spend their entire lifecycle on certain groups of host plants. Aphid species that apparently have wide host plant ranges have often turned out to be complexes of host-specialized biotypes. Here we investigated the various host-associated populations of the two recently diverged species, *Aphis gossypii* and *A. rhamnicola*, having multiple primary hosts, to understand the complex evolution with host-associated speciation. Using mitochondrial DNA marker and nine microsatellite loci, we reconstructed the haplotype network, and analyzed the genetic structure and relationships. Approximate Bayesian computation was also used to infer the ancestral primary host and host-associated divergence, which resulted in *Rhamnus* being the most ancestral host for *A. gossypii* and *A. rhamnicola*. As a result, *Aphis gossypii* and *A. rhamnicola* do not randomly use their primary and secondary host plants; rather, certain biotypes use only some secondary and specific primary hosts. Some biotypes are possibly in a diverging state through specialization to specific primary hosts. Our results also indicate that a new heteroecious race can commonly be derived from the heteroecious ancestor, showing strong evidence of ecological specialization through a primary host shift in both *A. gossypii* and *A. rhamnicola*. Interestingly, *A. gossypii* and *A. rhamnicola* shared *COI* haplotypes with each other, thus there is a possibility of introgression by hybridization between them by cross-sharing same primary hosts. Our results contribute to a new perspective in the study of aphid evolution by identifying complex evolutionary trends in the *gossypii* sensu lato complex.

## Introduction

Phytophagous insects are a group of tremendous diversity that covers a quarter of all known terrestrial biodiversity [1,2]. It has long been a concern to identify the evolutionary force of

Invasive Species Management Program funded
this study via a grant (2018002270005) awarded to
HK. (www.keiti.re.kr) The Basic Science Research
Program through the National Research
Foundation of Korea (NRF) funded by the Ministry
of Education funded this study via a grant
(2018R1D1A3B07044298) awarded to HK. (www.
nrf.re.kr) The funders had no role in study design,
data collection and analysis, decision to publish, or
preparation of the manuscript. BTL Bio-Test Labor
GmbH provided support via salary for TT, but did
not have any additional role in the study design,
data collection and analysis, decision to publish, or
preparation of the manuscript. (www.biotestlab.de)
The specific roles of these authors are articulated in
the 'author contributions' section.

**Competing interests:** The authors have read the
journal's policy and declare the following
competing interests: TT is a paid employee of BTL
Bio-Test Labor GmbH (www.biotestlab.de). There
are no patents, products in development or
marketed products associated with this research to
declare. This does not alter our adherence to PLOS
ONE policies on sharing data and materials.

their remarkable diversity [3]. In most cases, phytophagous lineages have a much higher diversity than their closely related non-phytophagous lineages [3,4]. They have an intimate relationship with certain and non-random host plant groups [5]. These findings often lead to the assumption that the host plant relationship holds the key to diversification in phytophagous insects [5,6]. In particular, many observations of host-specific races have provided crucial evidence to support these assumptions [2,7,8]. Walsh [9] first proposed an ecological speciation scenario to explain the formation of sympatric host-associated populations (HAPs). The basic scenario of ecological speciation is that the transition to new host plants provides opportunities to have novel ecological niches for phytophagous insects, and contribute to different host preferences and genetic isolation; and this subsequently resulted in speciation [10].

The ecological speciation concept has also been extensively applied to explain the process of aphid speciation [11–14]. Aphids provide a good model system, since the majority of species are host-specific, and they spend their entire lifecycles on certain groups of host plants [15,16]. Over the past decades, numerous studies have focused on the host relationship as a major factor in their speciation [15]. As one of the decisive examples, aphid species that apparently have wide host plant ranges (i.e. polyphagous) have often turned out to be complexes of host-specialized biotypes [11,12]. It is well known that the pea aphid, *Acyrthosiphon pisum* (Harris), the most well-studied aphid species, is a set of genetically well-distinguished biotypes linked with different legume species, which can be described as an example of sympatric speciation [11,17–19]. A similar pattern is also found in *Uroleucon* spp., which live on certain species or some closely related plants within Asteraceae [20]. These groups of aphids show typical diversification patterns on a narrow range of related host plants (within the same family) through a trade-off in host use, gradual reduction of gene flow, and genetic drift [11,19]. In addition, these examples are only possible if there is a mass of communities between closely related plants, which are mainly reported when many species of host plants live in similar conditions, such as asters and legumes [11,20,21].

In contrast to these classic examples, several species exhibit extreme polyphagous behavior, which occurs on a wide variety of unrelated plant families [15,22]. As one of the most representative polyphagous aphids, the cotton-melon aphid, *Aphis gossypii* Glover, is associated with about 900 plants belonging to 116 plant families, including more than 100 important crops worldwide [22,23]. The lifecycle of *A. gossypii* is as highly variable as its wide distribution range [22,24,25]. It has long been described as permanently anholocyclic [26], which is why studies on host change and primary hosts have rarely been conducted. First, Kring [27] reported that this aphid can perform a holocycly in North America. Today we have to assume that *gossypii* occurs in North America, East Asia, and Europe in numerous lines, some of which have a permanent anholocyclic reproduction, while others also have a holocyclic generation cycle [28]. Those points aside, the most unusual feature of this species is that they use primary hosts belonging to various unrelated plant families (e.g. Malvaceae, Punicaceae, Rhamnaceae, Rubiaceae and Rutaceae) [22,26]. This is particularly interesting in evolutionary terms, and makes *A. gossypii* a good model for understanding the evolutionary process associated with the primary hosts of heteroecious aphids.

Approximately 10% of 5,000 aphid species exhibit the seasonal host alternation (i.e. heteroecy) between primary and secondary hosts, which mysteriously are comprised with a set of phylogenetically unrelated host plants [22,29,30]. In addition, among all phytophagous insects, the complex life cycle completed by multiple generations is known to be limited to the aphids (Aphidoidea) [31,32]. In particular, the success of species diversity and remarkable host plant relationship are believed to be attributed to multiple acquisition and the loss of heteroecy [33,34]. Lifecycles of heteroecious aphids usually comprise migration from the primary host to the secondary host [35]. For one cycle, sexual and asexual reproductions occur accompanied

by several morphological changes as the generation passes [35]. Most heteroecious aphids use a much narrower range of primary host (within the same family or genus), even if they have a wide range of secondary hosts [36]. These patterns are even observed to extend to the closely related aphid species. For example, eight *Hyperomyzus* species are known to have host alternation with the primary hosts within the genus *Ribes*, even though these species have wider relationships with various secondary hosts, such as Asteraceae and Scrophulariaceae [22].

There are different views about these contrasting host ranges of primary and secondary hosts. The first view suggested that primary host specialization is in evolutionary terms more favored, because a primary host not only provides nutrients, but also a mating place [37]. For heteroecious aphids, a primary host is a place where sexual reproduction takes place, as well as the overwintering eggs hatch, and the following fundatrix stage [35,38]. However, on the secondary host, only asexual reproduction occurs before the migration for overwintering [35,38]. Thus, having different primary hosts, rather than those with different secondary hosts, may have a greater significance in reproduction. In other words, a constraint of primary host is possibly linked to the mating success of a species [15,37]. The second view hypothesized that host alternation is only a by-product of an evolutionary process that occurs due to the phylogenetic constraints of fundatrix to the primary host [32,33]. This is the so-called fundatrix specialization hypothesis (Fig 1), which is based on the following assumptions: i) monoecy is evolutionarily more favored than heteroecy, ii) primary hosts are more ancestral than secondary hosts, iii) the fundatrix is highly adaptive to the primary host, but maladaptive to the secondary host, and iv) secondary hosts are more labile, and more recently obtained than primary hosts [31,32,39]. Under this hypothesis, the loss of a primary host was described as escape, and specialization to a specific secondary host was believed to be the only evolutionary way [31–33,39,40].

Having multiple primary hosts in *A. gossypii* [22,24] suggests the possibility of host-associated speciation. Indeed, previous population genetic studies have shown that *A. gossypii* is a complex of several genetically distinct biotypes associated with some plant families (e.g. Cucurbitaceae, Malvaceae, and Solanaceae) [12,41]. However, these studies only targeted anholocyclic lineages of *A. gossypii* collected from certain crops, and only a few primary hosts

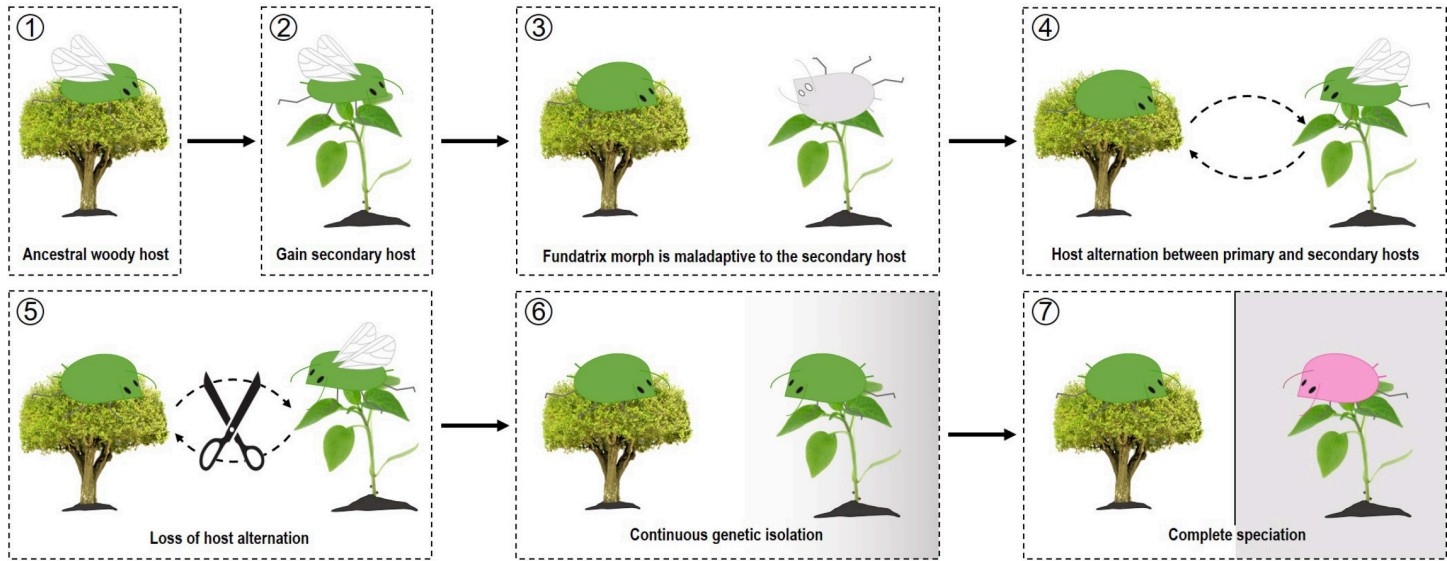

① Ancestral woody host
② Gain secondary host
③ Fundatrix morph is maladaptive to the secondary host
④ Host alternation between primary and secondary hosts
⑤ Loss of host alternation
⑥ Continuous genetic isolation
⑦ Complete speciation

**Fig 1. A hypothetical scenario of the fundatrix specialization.** Concept of speciation by loss of primary host from Moran [33].

(e.g. *Hibiscus syriacus* and *Punica granatum*) [12,41]. They live on a much broader range of wild plants, and the primary hosts are also very diverse. Nevertheless, we know surprisingly little about the primary host-associated genetic structure in this species. In particular, there is no study on the genetic structure between primary and secondary HAPs of *A. gossypii*. Therefore, to better understand the evolutionary trends in *A. gossypii*, further genetic analyses encompassing wild HAPs are needed.

In addition, confirming the ancestral host plant is crucial to understanding the evolutionary process of aphids. Among several host plants used as the primary host of *A. gossypii*, the genus *Rhamnus* (incl. *Frangula*) in Rhamnaceae is the most strongly presumed to be a ancestral host of the *gossypii* sensu lato complex group [42,43]. There are several reasons why *Rhamnus* is regarded as a ancestral host. The first reason is that most species belonging to the *gossypii* group show congruent use of primary hosts in *Rhamnus* [44], while the second reason is the possibility that the *gossypii* complex group and *Rhamnus* chronologically co-evolved based on molecular dating and fossil record [44,45]. Because of this, *Rhamnus* has been believed to be at the center of the host-associated evolution in the *gossypii* complex group. Nevertheless, the relationships between *Rhamnus* and other secondary HAPs have not yet been investigated.

This study aims to investigate the evolutionary trends of the two closely-related host-alternating species, *A. gossypii* and *A. rhamnicola*, based on population genetic analyses of various primary and secondary HAPs. *Aphis gossypii* shows a typical heteroecious holocyclic lifecycle in Korea, for which various perennials and woody plants are known to be used as primary hosts [22], even though several anholocyclic isolates have been found in the secondary hosts [12]. *Aphis rhamnicola* is a recently described cryptic species of *A. gossypii* that shares *Rhamnus* spp. as primary hosts, but has a somewhat different range of secondary hosts [46]. We conducted population genetic analyses of the two species in a comprehensive set of populations from primary and secondary host plants that were mostly collected from South Korea (except for one population from the UK). We used two molecular approaches in this study. First, reconstructing the haplotype network based on *COI* barcode, we confirmed their speciation pattern and genetic relationships of the aphid HAPs specialized on various host plants. Second, using nine microsatellite loci, we analyzed the genetic structure to identify the relationships between the HAPs of the two species, and to clarify the host shifting or switching process between the primary and secondary hosts. We also inferred the most likely ancestral host for the primary HAPs, which could be strongly suggested through the results of this study, by using approximate Bayesian computation methods.

## Materials and methods

### Taxon sampling and DNA extraction

As all collections have not been carried out in restricted areas, national parks, etc. where permits are required, it is clearly stated that there is no content regarding collection permits. To examine the genetic structure, diversity and host-associated evolution between primary and secondary HAPs, we used 578 individual aphid samples of 36 HAPs, selectively pooled from 116 different collections, within the two species, *A. gossypii* and *A. rhamnicola*, which were collected from 36 different host plants—perennial, annual, and biannual; woody and herbaceous —in 16 plant families (Table 1). For forthcoming analyses, primary hosts and secondary hosts were defined based on the following criteria: i) The obvious primary hosts are plants that have collected sexuparae and fundatrix morphs. In addition to the obvious primary host, we also considered plants meeting the following two conditions as primary hosts. ii) Plants previously recorded as primary hosts of *A. gossypii* with reference to Inaizumi [23,25] and Blackman and Eastop [28] or iii) the case when the collecting time is early spring (April-May) or late autumn

**Table 1. Summary statistics for microsatellite data from all aphid populations.**

| Pop. ID | No. | Sorted lineage | Host plant | Host type[d] | MLGs | $N_A$ | $H_S$ | $R_S$ | Ho (±s.e.) | He (±s.e.) | HWE[e] | FIS[f] |
|---|---|---|---|---|---|---|---|---|---|---|---|---|
| Ag_IL | 10 | *Aphis gossypii* Group 2 | *Ilex cornuta* | P, W | 5 | 2.33 | 0.36 | 2.12 | 0.49 (0.15) | 0.37 (0.10) | ns | -0.35 |
| Ag_CU | 20 | *Aphis gossypii* Group 2 | *Cucumis sativus* | A, H | 16 | 3.11 | 0.43 | 2.43 | 0.49 (0.11) | 0.43 (0.08) | ns | -0.16 |
| Ag_CM | 30 | *Aphis gossypii* Group 2 | *Cucurbita moschata* | A, H | 22 | 4.56 | 0.46 | 2.80 | 0.46 (0.11) | 0.46 (0.08) | ns | 0.00 |
| Ag_KA | 8 | *Aphis gossypii* Group 2 | *Kalanchoe daigremontiana* | P, W | 6 | 2.56 | 0.50 | 2.48 | 0.68 (0.14) | 0.51 (0.08) | ns | -0.35 |
| Ag_SO | 20 | *Aphis gossypii* Group 2 | *Solanum melongena* | P, W | 10 | 4.56 | 0.50 | 3.04 | 0.53 (0.13) | 0.50 (0.11) | ns | -0.07 |
| Ag_CA | 25 | *Aphis gossypii* Group 2 | *Capsicum annuum* | P, W | 6 | 2.44 | 0.39 | 2.12 | 0.64 (0.16) | 0.40 (0.10) | *excess | -0.63 |
| Ag_CP | 5 | *Aphis gossypii* Group 2 | *Capsicum annuum* var. *angulosum* | P, W | 4 | 1.89 | 0.36 | 1.89 | 0.64 (0.16) | 0.39 (0.10) | *excess | -0.79 |
| Ag_PU | 27 | *Aphis gossypii* Group 1 | *Punica granatum*[c] | P, W | 25 | 5.56 | 0.56 | 3.41 | 0.57 (0.08) | 0.56 (0.08) | ns | -0.01 |
| Ag_EL | 10 | *Aphis gossypii* Group 1 | *Eleanterococcus senticosus* | P, W | 9 | 2.67 | 0.43 | 2.40 | 0.42 (0.11) | 0.43 (0.09) | ns | 0.03 |
| Ag_HI | 60 | *Aphis gossypii* Group 1 | *Hibiscus syriacus*[c] | P, W | 59 | 7.33 | 0.50 | 3.25 | 0.49 (0.09) | 0.50 (0.08) | ns | 0.03 |
| Ag_HR | 10 | *Aphis gossypii* Group 1 | *Hibiscus rosa-sinensis*[c] | P, W | 8 | 2.22 | 0.31 | 2.02 | 0.38 (0.12) | 0.31 (0.09) | ns | -0.23 |
| Ag_EU | 10 | *Aphis gossypii* Group 1 | *Euonymus trapococca* | P, W | 9 | 3.56 | 0.52 | 3.04 | 0.56 (0.11) | 0.52 (0.10) | ns | -0.07 |
| Ag_EJ | 20 | *Aphis gossypii* Group 1 | *Euonymus japonicas* | P, W | 17 | 4.56 | 0.52 | 3.20 | 0.58 (0.10) | 0.52 (0.09) | ns | -0.12 |
| Ag_CI | 20 | *Aphis gossypii* Group 1 | *Citrus unshiu* | P, W | 16 | 4.89 | 0.51 | 3.17 | 0.57 (0.10) | 0.52 (0.06) | ns | -0.11 |
| Ag_FO | 10 | *Aphis gossypii* Group 1 | *Forsythia koreana* | P, W | 10 | 3.44 | 0.47 | 2.86 | 0.50 (0.11) | 0.47 (0.10) | ns | -0.07 |
| Ag_CE | 20 | *Aphis gossypii* Group 1 | *Celastrus orbiculatus*[c] | P, W | 19 | 4.44 | 0.46 | 2.90 | 0.52 (0.11) | 0.47 (0.09) | ns | -0.13 |
| Ag_ER | 10 | *Aphis gossypii* Group 1 | *Erigeron annuus* | A, H | 8 | 2.78 | 0.37 | 2.31 | 0.46 (0.12) | 0.37 (0.09) | ns | -0.23 |
| Ag_SN | 8 | *Aphis gossypii* Group 1 | *Sonchus oleraceus* | B, H | 7 | 3.00 | 0.45 | 2.59 | 0.50 (0.11) | 0.45 (0.08) | ns | -0.12 |
| Ag_CO | 18 | *Aphis gossypii* Group 1 | *Cosmos bipinnatus* | A, H | 16 | 3.56 | 0.51 | 2.82 | 0.48 (0.10) | 0.50 (0.08) | ns | 0.05 |
| Ag_CL | 10 | *Aphis gossypii* Group 1 | *Clinopodium chinense* var. *parviflorum* | P, H | 4 | 1.67 | 0.20 | 1.55 | 0.28 (0.11) | 0.21 (0.08) | ns | -0.37 |
| Ag_CT | 10 | *Aphis gossypii* Group 1 | *Catalpa ovata*[c] | P, W | 8 | 1.78 | 0.30 | 1.70 | 0.38 (0.11) | 0.30 (0.08) | ns | -0.26 |
| Ag_CJ | 10 | *Aphis gossypii* Group 1 | *Callicarpa japonica* | P, W | 8 | 3.11 | 0.48 | 2.64 | 0.30 (0.10) | 0.47 (0.08) | *deficit | 0.38 |
| Ag_RH | 20 | *Aphis gossypii* Group 1 | *Rhamnus davurica*[c] | P, W | 20 | 8.56 | 0.70 | 4.62 | 0.79 (0.07) | 0.70 (0.07) | ns | -0.12 |
| Ar_SE | 10 | *Aphis rhamnicola* Group 1[a] | *Sedum kamtschaticum* | P, H | 8 | 2.67 | 0.43 | 2.38 | 0.43 (0.10) | 0.43 (0.07) | ns | -0.01 |
| Ar_PE | 10 | *Aphis rhamnicola* Group 1[a] | *Perilla frutescens* var. *frutescens* | A, H | 6 | 1.89 | 0.27 | 1.77 | 0.42 (0.15) | 0.28 (0.09) | *excess | -0.54 |
| Ar_YO | 11 | *Aphis rhamnicola* Group 3[b] | *Youngia sonchifolia* | B, H | 10 | 3.67 | 0.47 | 2.90 | 0.23 (0.07) | 0.46 (0.10) | *deficit | 0.51 |
| Ar_IX | 11 | *Aphis rhamnicola* Group 3[b] | *Ixeris strigose* | P, H | 9 | 2.89 | 0.39 | 2.39 | 0.27 (0.08) | 0.38 (0.09) | ns | 0.30 |
| Ar_RH | 8 | *Aphis rhamnicola* Group 1 | *Rhamnus davurica*[c] | P, W | 8 | 3.44 | 0.55 | 3.02 | 0.56 (0.07) | 0.55 (0.04) | ns | -0.01 |
| Ar_CO | 30 | *Aphis rhamnicola* Group 1 | *Commelina communis* | A, H | 30 | 6.67 | 0.59 | 3.58 | 0.55 (0.09) | 0.59 (0.08) | ns | 0.07 |
| Ar_LE | 10 | *Aphis rhamnicola* Group 1 | *Leonurus japonicus* | B, H | 10 | 4.11 | 0.53 | 3.07 | 0.61 (0.10) | 0.53 (0.07) | ns | -0.16 |
| Ar_PH | 7 | *Aphis rhamnicola* Group 1 | *Phryma leptostachy* | P, H | 5 | 2.33 | 0.43 | 2.23 | 0.19 (0.08) | 0.41 (0.06) | *deficit | 0.55 |
| Ar_ST | 6 | *Aphis rhamnicola* Group 2 | *Stellaria media* | B, H | 5 | 1.89 | 0.29 | 1.83 | 0.39 (0.14) | 0.30 (0.09) | ns | -0.36 |
| Ar_LY | 10 | *Aphis rhamnicola* Group 2 | *Lysimachia coreana* | P, H | 10 | 3.56 | 0.49 | 2.92 | 0.49 (0.12) | 0.49 (0.10) | ns | 0.01 |
| Ar_CB | 24 | *Aphis rhamnicola* Group 2 | *Capsella bursa-pastoris* | B, H | 24 | 4.11 | 0.50 | 2.83 | 0.55 (0.13) | 0.50 (0.10) | ns | -0.10 |
| Ar_VE | 10 | *Aphis rhamnicola* Group 2 | *Veronica insularis* | P, H | 10 | 3.22 | 0.49 | 2.79 | 0.53 (0.14) | 0.49 (0.10) | ns | -0.10 |
| Ar_RU | 40 | *Aphis rhamnicola* Group 2 | *Rubia akane* | P, W | 32 | 5.56 | 0.49 | 2.87 | 0.36 (0.08) | 0.49 (0.07) | *deficit | 0.27 |

Number of multilocus genotypes (MLGs); observed heterozygosity (Ho); expected heterozygosity (He); Hardy-Weinberg Equilibrium (HWE); gene diversity ($H_S$); mean number of alleles ($N_A$); allelic richness ($R_S$). ns: Non-significance in HWE (P > 0.05).

*P values for heterozygote deficit or heterozygote excess. (P < 0.001)

[a] possibly other cryptic species A.

[b] possibly other cryptic species B.

[c] known as primary host [25,28].

[d] Host type, P: Perennial, A: Annual, B: Biennial or annual, W: Woody, H: Herbaceous.

[e] HWE estimated excluding the clonal copies of MLGs

[f] FIS multiple loci.

(October-November) based on the lifecycle of *A. gossypii* on the Korean Peninsula; However, even if these two conditions were met, annual or biennial plants were excluded from the primary host. It was also not considered as the primary host if aphid collected in a greenhouse. As a consequence, all remaining plants not falling under the above conditions were considered as the secondary hosts. In our study, the host-associated population (HAP) means a collective population pooled from several temporally and/or geographically different collections in the same plant species (S1 Table). In *A. gossypii* with a large spectrum of host utilization, 25 HAPs were collected from its various primary and secondary hosts (S1 Table). As *A. rhamnicola* was recently recorded found in *Rhamnus* spp., sharing and co-existing with *A. gossypii* in *Rhamnus* as a primary host [46], nine HAPs of *A. rhamnicola* were also collected from its various primary and secondary hosts (S1 Table).

These collections were acquired from South Korea, except for those of *A. gossypii* from *Catalpa ovata* in the UK (S1 Table). To avoid the chance of sampling individuals from the same parthenogenetic colony, each aphid was collected from a different host plant, or a different isolated colony. All of the fresh aphid specimens used for molecular analyses were collected and preserved in (95 or 99) % ethanol, and stored at -70°C. Total genomic DNA was extracted from single individuals using a DNeasy® Blood & Tissue Kit (QIAGEN, Inc., Dusseldorf). To preserve voucher specimens from the DNA extracted samples, we used a non-destructive DNA extraction protocol [43]. The entire body of the aphid was left in the lysis buffer with protease K solution at 55°C for 24 h, and the cleared cuticle dehydrated.

## Species lineage sorting

In some aphid groups, morphological identification can be ambiguous, due to the lack of conclusive morphological evidence. The *Aphis* group is one of the most typical groups with the above problem. As a complementary way to avoid misidentification, host plant relationships, morphologies, and molecular tools are widely used to identify aphids [43,46–48]. Because our study aims to demonstrate intra-specific genetic relationships based on host plant associations, species lineage sorting is significant to prevent biases of the results. The two *Aphis* species, *A. gossypii* and *A. rhamnicola*, we study here are not only very similar in morphology, but also share several host plants due to the polyphagy. Although we performed species identification through morphology and host plant relationships as a first step and also tested DNA barcoding for all individuals collected on their shared host plants (e.g. *Capsella*, *Rhamnus*, and *Rubia*), we found that there were a lot of the haplotypes cross-shared between *A. gossypii* and *A. rhamnicola* (see Results). Therefore, instead of identifying the species with 36 HAPs, we applied the dominant assignment (white, green, blue, red, dark blue) of the genetic structure ($K$ = 3, 4, 5) by STRUCTURE as well as the PCoA results (see Results) to sort their lineags into five groups as *Aphis gossypii* Group 1, *A. g.* Group 2, *A. rhamnicola* Group 1, *A. r.* Group 2 and *A. r.* Group 3 (Table 1). Accordingly, '*Aphis gossypii*' and '*A. rhamincola*', which are mentioned later, are meant to include all group lineages containing the HAPs assigned by the results. S1 Table shows detailed information for lineage sorted samples used in DNA analyses.

## Haplotype analysis

A 658 bp of the partial 5' region of the *cytochrome c oxidase subunit I* gene (*COI*), namely *COI* DNA barcode [49], was amplified using the universal primer sets: LEP-F1 5'-ATTCAACCAAT CATAAAGATAT-3' and LEP-R1, 5'-TAAACTTCTGGATGTCCAAAAA-3'. A polymerase chain reaction (PCR) was performed with AccuPower® PCR Premix (Bioneer, Daejeon, Rep. of Korea) in 20 mL reaction mixtures under the following conditions: initial denaturation at 95°C for 5 min; followed by 35 cycles at 94°C for 30 s, an annealing temperature of 45.2°C for

40 s, an extension at 72˚C for 45 s, and the final extension at 72˚C for 5 min. All PCR products were assessed using a 1.5% agarose gel electrophoresis. Successfully amplified samples were purified using a QIAquick PCR purification kit (Qiagen, Inc.), and then immediately sequenced using an automated sequencer (ABI Prism 3730XL DNA Analyzer) at Bionics Inc. (Seoul, Korea). Both morphological identification, based on voucher specimens in the insect museum in Kunsan National University with descriptions of Blackman and Eastop [22], Lee and Kim [24], Heie [26], and molecular identification method using the *COI* DNA barcode region for comparison with the previous *COI* DNA barcode database, were used [43,46,50].

All sequences that were obtained for DNA barcoding were initially examined and assembled using CHROMAS 2.4.4 (Technelysium Pty Ltd., Tewantin, Qld, AU) and SEQMAN PRO ver. 7.1.0 (DNA Star, Inc., Madison, Wisconsin, USA). In this step, poor-quality sequences were discarded to avoid biases. The final dataset containing 187 sequences was aligned using MAFFT ver. 7 [51], an online utility. Some ambiguous front and back sequences were removed at this stage, resulting in sequences of 583 bp that were finally used for haplotype analysis. All sequences were deposited in GenBank (accession no. MT461429-MT461602). The *COI* haplotypes of *A. gossypii* complex were analyzed using DNASP ver. 6.12.03 [52]. A median-joining network (MJ) was built using NETWORK ver. 5.0.1.1 [53]. The MJ result was annotated with host plants or species, and then visually summarized in Fig 2.

### Microsatellite genotyping

In this study, all 578 individuals of four species were successfully genotyped using nine microsatellite loci (AGL1-2, AGL1-10, AGL1-11, AGL1-15, AGL1-16, AGL1-20, AGL1-21, AGL1-22, and AGL2-3b) previously isolated from the soybean aphid [55]. In the preliminary study, we had already checked the cross-species amplification test of these loci on *A. fabae*, *Hyalopterus pruni*, *Rhopalosiphum padi*, and *Schizaphis graminum*, as well as *A. gossypii* in the tribe Aphidini. There were the previously developed loci from *A. gossypii* [56], but we used the nine loci developed from *A. glycines* [55], because we noticed that the polymorphism of the latter was higher than that of the former, which was advantageous to amplify the loci between different species. In the aphid group, several studies showed that microsatellite loci were available between related species within the aphid family as a utility of cross-species amplification [57,58].

Microsatellite amplifications were performed using GeneAll® *Taq* DNA Polymerase Premix (GeneAll, Seoul, Korea) in 20 μL reaction mixtures containing 0.5 μM forward labeled with a fluorescent dye (6-FAM, HEX, or NED) & reverse primers, and 0.05 μg of DNA template. PCR was performed using a GS482 thermo-cycler (Gene Technologies, Essex), according to the following procedure: initial denaturation at 95˚C for 5 min, followed by 34 cycles of 95˚C for 30 s; annealing at 56˚C for 40 s; extension at 72˚C for 45 s, and a final extension at 72˚C for 5 min. PCR products were visualized by electrophoresis on a 1.5% agarose gel with a low-range DNA ladder to check for positive amplifications. Automated fluorescent fragment analyses were performed on the ABI PRISM 377 Genetic Analyzer (Applied Biosystems), and allele sizes of PCR products were calibrated using the molecular size marker, ROX labeled-size standard (GenScan™ ROX 500, Applied Biosystems). Raw data on each fluorescent DNA product was analyzed using GeneMapper® version 4.0 (Applied Biosystems).

### Microsatellite data analysis

We used GENALEX 6.503 [59] to identify multilocus genotypes (MLGs) among populations. The program FSTAT 2.9.3.2 [60] was used to estimate the mean number of alleles ($N_A$), gene diversity ($H_S$), and allelic richness ($R_S$). Observed ($H_O$) and expected heterozygosity ($H_E$)

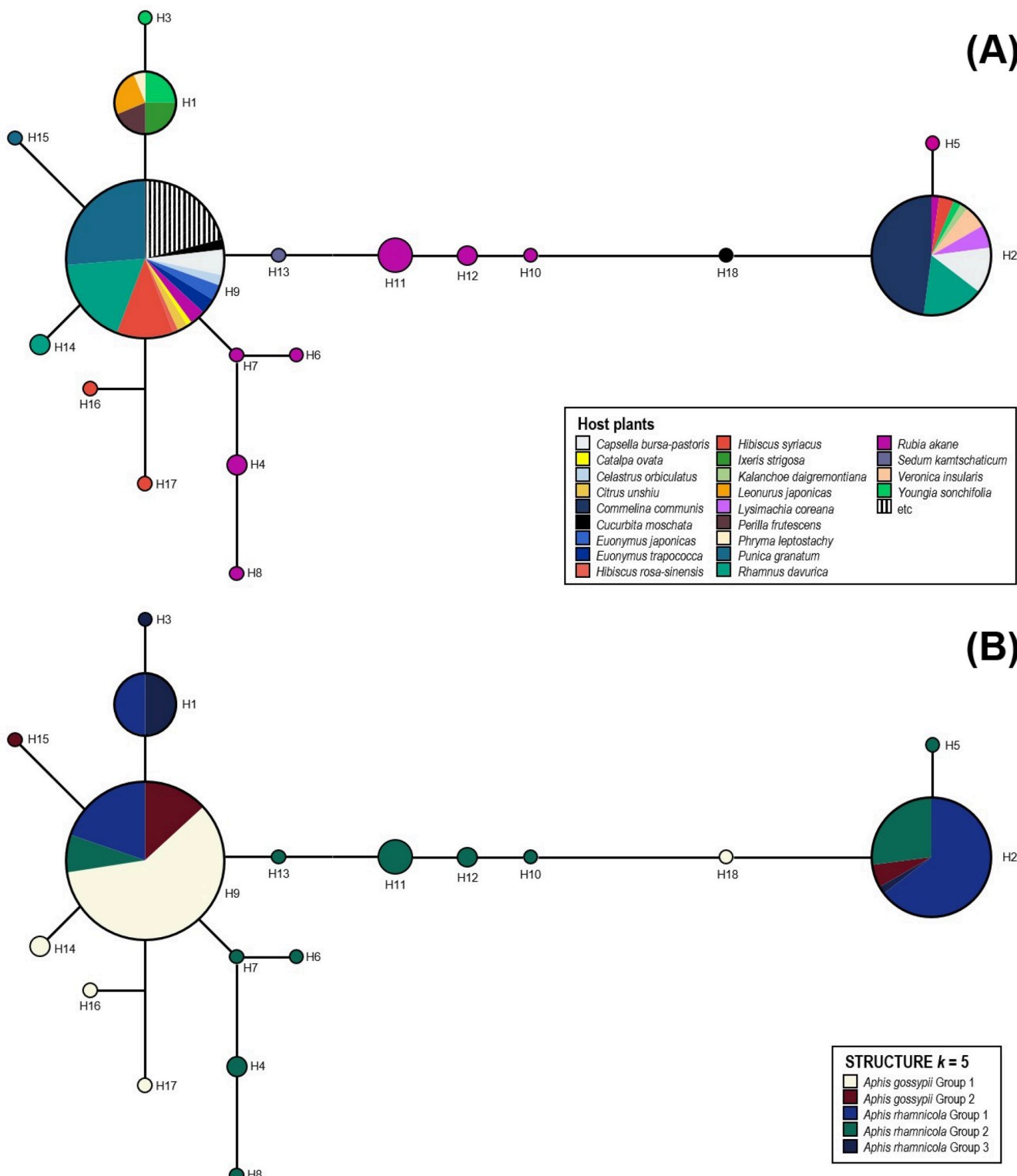

**Fig 2.** (A) Haplotype network for *COI* DNA barcode dataset (583 bp) using NETWORK ver. 5.0.1.1 [54]. Pie chart distribution based on each HAP; (B) Haplotype network for *COI* DNA barcode dataset (583 bp) using NETWORK. Pie chart distribution based on each group of five lineages in the two species.

values among loci were estimated using GENEPOP 4.0.7 [61] among the population data (HAPs) sets. Levels of significance for Hardy–Weinberg equilibrium (HWE) and linkage disequilibrium tests were adjusted using the sequential Bonferroni correction for all tests involving multiple comparisons [62]. Deviations from HWE were tested for heterozygote deficiency or excess. Because the clonal copies of MLGs due to the parthenogenetic life cycle of aphids could affect and distort the estimation of HWE [63], we used a reduced data set containing only one copy of each MLG when estimating HWE. Several assumptions of HWE still can be violated, thereby these estimates are used only for descriptive purposes even although the clonal MLG copies were removed from data analysis [63]. MICRO-CHECKER [64] was used to test for null alleles [65] and identify possible scoring errors, because of the large-allele dropout and stuttering.

We used ARLEQUIN 3.5.1.2 [66] for calculations of pairwise genetic differentiation ($F_{ST}$) values [67], in which populations were assigned by 36 HAPs of the two species. The statistical significance of each value was assessed by computing the pairwise comparison of the observed value in 100,000 permutations. Groupings based on three different cases, (1) *gossypii* vs *rhamnicola*, (2) perennial vs non-perennial host groups in *A. gossypii*, (3) perennial vs non-perennial host groups in *A. rhamnicola*, were tested independently with analysis of molecular variance [AMOVA; 68] in ARLEQUIN, with significance determined using the nonparametric permutation approach described by Excoffier et al. [69].

To examine the genetic relationships between 578 individual samples of four species, principal coordinate analysis (PCoA), also in GENALEX [59], further explored population relationships using the microsatellite loci, making no *a priori* assumptions about population groupings. Codominant genotypic genetic distance was calculated to make tri-matrix of pairwise populations, and then each population plot was created with coordinates based on the first two axes.

The program STRUCTURE 2.3.3 [70] was used to test for the existence of population structuring among all samples, by estimating the number of distinct populations (*K*) present in the set of samples, using a Bayesian clustering approach. We assessed likelihoods for models with the number of clusters ranging *K* = (1 to 15). The length of the initial burn-in period was set to 100,000 iterations, followed by a run of 1,000,000 Markov chain Monte Carlo (MCMC) repetitions, of which the analysis was replicated 10 times, to ensure convergence on parameters and likelihood values. Parameter sets of ancestry, allele frequency, and advanced models remained as defaults. Following the method of Evanno et al. [71], we calculated Δ*K* based on the second-order rate of change in the log probability of data with respect to the number of population clusters from the STRUCTURE analysis. To determine the correct value of *K*, both the likelihood distribution being to plateau or decrease [70] and the peak value of the Δ*K* statistic of Evanno et al. [71] was estimated. The single run at each *K* yielding the highest likelihood of the data given the parameter values was used for plotting the distributions of individual membership coefficients (*Q*) with the program DISTRUCT [72].

We performed assignment tests using GENECLASS 2 [73], in which populations were assigned to 36 HAPs of the two species. For each individual of a population, the program calculates the probability of belonging to any other reference population, or of being a resident of the population where it was sampled. The sample with the highest probability of assignment was considered the most likely source for the assigned genotype. In this study, we checked the mean assignment rate from 391 *A. gossypii* or 187 *A. rhamnicola* individuals into each population (source), to confirm the possible origin of each HAP. We used a Bayesian method of estimating population allele frequencies [74]. Monte Carlo re-sampling computation (100,000 simulated individuals) was used to infer the significance of assignments (alpha = 0.01).

## Approximate Bayesian computation analysis

To estimate the relative likelihood of the most likely ancestral HAP of *A. gossypii*, an approximate Bayesian computation (ABC) was performed for the microsatellite dataset as implemented in DIYABC version 1.0.4 [75]. DIYABC allows the comparison of complex scenarios involving bottlenecks, serial or independent introductions, and genetic admixture events in introduced populations [76]. The parameters for modeling scenarios are the times of split or admixture events, the stable effective population size, the effective number of founders in introduced populations, the duration of the bottleneck during colonization, and the rate of admixture [77]. The software generates a simulated data set used to estimate the posterior distribution of parameters, in order to select the most likelihood scenario [77]. DIYABC generates a simulated data set that is then used to select those most similar to the observed data set, and the so-called selected data set ($n_\delta$), which are finally used to estimate the posterior distribution of parameters [75]. Recently, this ABC software package has been widely used, such as for inferring the demographic history of populations and species [78,79], and testing potential bottleneck events [80].

To infer the most likely ancestral primary host of *A. gossypii*, among the whole microsatellite dataset, we tested three different ABC analyses using the original or partial dataset. We hypothesized the evolutionary scenarios following our results obtained from the *COI* haplotype network and two Bayesian analyses, STRUCTURE and GENECLASS2 (see "Results"). The previous studies have already revealed that *A. rhamnicola* was located in the more ancestral position within the phylogeny of the *A. gossypii* group [43,44], which population was therefore set to the most ancestral position on the genealogy of the two ABC tests.

In the first analysis, based on the result of STRUCTURE (*K* = 3), we compared eight evolutionary scenarios (A1–A8) using a dataset that included 578 individuals from four population groups, which consisted of 75 individuals from the 'BLUE' group (Ar_SE, Ar_PE, Ar_IX, Ar_YO, Ar_CO, Ar_PH, Ar_RH, Ar_LE), 90 from the 'GREEN' group (Ar_ST, Ar_VE, Ar_LY, Ar_CB, Ar_RU), 30 from the 'MIXBW (BLUE+WHITE)' group (Ag_RH, Ag_CJ), and 361 from the 'WHITE' group (Ag-IL, Ag_CE, Ag_EU, Ag_EJ, Ag_PU, Ag_CU, Ag_CM, Ag_KA, Ag_EL, Ag_HI, Ag_HR, Ag_FO, Ag_CI, Ag_ER, Ag_SN, Ag_CO, Ag_SO, Ag_CA, Ag_CP, Ag_CL, Ag_CT) (S1 Fig). Scenario A1 considered (1) GREEN originated from BLUE, (2) MIXBW subsequently originated from GREEN, and then (3) WHITE originated from MIXBW. Scenario A2 considered (1) WHITE originated from BLUE, (2) MIXBW subsequently originated from WHITE, and then (3) GREEN originated from MIXBW. Scenario A3 considered (1) BLUE originated from GREEN, (2) MIXBW subsequently originated from BLUE, and then (3) WHITE originated from MIXBW. Scenario A4 considered (1) WHITE originated from MIXBW, (2) BLUE subsequently originated from WHITE, and then (3) GREEN originated from BLUE. Scenario A5 considered (1) GREEN originated from MIXBW, (2) BLUE subsequently originated from GREEN, and then (3) WHITE originated from BLUE. Scenario A6 considered (1) GREEN originated from WHITE, (2) MIXBW subsequently originated from GREEN, and then (3) BLUE originated from MIXBW. Scenario A7 considered (1) BLUE originated from MIXBW, (2) GREEN subsequently originated from BLUE, and then (3) WHITE originated from GREEN. Scenario A8 considered (1) BLUE originated from WHITE, (2) MIXBW subsequently originated from BLUE, and then (3) GREEN originated from MIXBW.

In the second analysis, based on the result of STRUCTURE (*K* = 4), we compared six evolutionary scenarios (B1–B6) using a dataset that included 311 individuals from four population groups, which consisted of 75 individuals from the 'BLUE' group (Ar_CO, Ar_PH, Ar_RH, Ar_SE, Ar_PE, Ar_LE), 90 from the 'GREEN' group (Ar_ST, Ar_VE, Ar_LY, Ar_CB, Ar_RU),

60 from the 'RED' group (Ag_IL, Ag_CU, Ag_CA, Ag_CP), and 86 from the 'WHITE' group (Ag_CE, Ag_FO, Ag_ER, Ag_SN, Ag_CO, Ag_CL, Ag_CT) (S2 Fig). Scenario B1 considered (1) GREEN originated from BLUE, (2) WHITE subsequently originated from GREEN, and then (3) RED originated from WHITE. Scenario B2 was basically similar to scenario B1, except for (1) WHITE originated from RED. Scenario B3 considered (1) WHITE originated from RED, (2) GREEN subsequently originated from WHITE, and then (3) BLUE originated from GREEN. Scenario B4 considered (1) GREEN originated from BLUE, (2) WHITE formerly originated from GREEN, and then (3) RED later originated from GREEN. Scenario B5 considered (1) GREEN originated from BLUE, (2) RED formerly originated from GREEN, and then (3) WHITE later originated from GREEN. Scenario B6 considered (1) WHITE originated from RED, (2) GREEN formerly originated from WHITE, and then (3) BLUE later originated from GREEN.

In the third analysis, based on the result of STRUCTURE (*K* = 4), we compared six evolutionary scenarios (C1–C6) using a dataset including 391 individuals from four population groups, except for BLUE and GREEN groups in the first and second analysis, which consisted of 30 individuals from the 'MBW (BLUE+WHITE)' group (Ag_RH, Ag_CJ), 207 from the 'MRW (RED+WHITE)' group (Ag_EU, Ag_EJ, Ag_PU, Ag_SO, Ag_CM, Ag_EL, Ag_HI, Ag_HR, Ag_CI), 68 from the 'RED' group (Ag_IL, Ag_CU, Ag_KA, Ag_CA, Ag_CP), and 86 from the 'WHITE' group (Ag_CE, Ag_FO, Ag_ER, Ag_SN, Ag_CO, Ag_CL, Ag_CT) (S3 Fig). Scenario C1 considered (1) MRW originated from MBW, (2) WHITE subsequently originated from MRW, and then (3) RED originated from WHITE. Scenario C2 was basically similar to scenario C1, except for (3) WHITE originated from RED. Scenario C3 considered (1) WHITE originated from MBW, (2) RED subsequently originated from WHITE, and then (3) MRW originated from RED. Scenario C4 was basically similar to scenario C3, except for (2) MRW subsequently originated from WHITE, and then (3) RED originated from MRW. Scenario C5 considered (1) RED originated from MBW, (2) WHITE subsequently originated from RED, and then (3) MRW originated from WHITE. Scenario C6 was basically similar to scenario C5, except for (3) WHITE originated from MRW.

We produced 1 000 000 simulated data sets for each scenario. We used a generalized stepwise model (GSM) as the mutational model for microsatellites, which assumes increases or reductions by single repeat units [75]. To identify the posterior probability (PP) of these three scenarios, the $n_\delta$ = 30 000 (1%) simulated datasets closest to the pseudo-observed dataset were selected for the logistic regression, which is similar to the $n_\delta$ = 300 (0.01%) ones for the direct approach [77]. The summary of statistics was calculated from the simulated and observed data for each of the tested scenarios, such as the mean number of alleles per locus (*A*), mean genetic diversity for each group and between group, genetic differentiation between pairwise groups ($F_{ST}$), classification index, shared alleles distance ($D_{AS}$), and Goldstein distance.

## Results

### Haplotype analysis

A total of 18 haplotypes were recognized from the 187 *COI* sequences of 36 host-associated populations of the two *Aphis* species (Fig 2). The most common haplotype was H9, followed by H2. Aphid samples from the three primary hosts: *Hibiscus*, *Rhamnus*, and *Rubia* were spread across these two major haplotypes (Fig 2A). All the samples from the remaining primary hosts (i.e. *Catalpa*, *Celastrua*, *Citrus*, *Euonymus*, and *Punica*) had H9 haplotype (Fig 2A). Unique haplotypes were mostly observed among the *Rubia* population (Fig 2A). *Cucurbita*, *Hibiscus*, *Punica*, *Rhamnus*, *Sedum*, and *Youngia* associated populations also had unique haplotypes. Among the secondary HAPs, samples from the *Capsella* were found in both H2 and

H9 haplotypes (Fig 2A). Samples from the *Youngia* was also observed in H1, H2, and H3 haplotypes (Fig 2A). However, the populations associated with the majority of secondary hosts only had one haplotype. H1 consisted of samples from secondary hosts, such as *Ixeris*, *Leonurus*, *Perilla*, *Phryma*, and *Youngia*. To compare *COI* haplotype and microsatellite genotype results, we overlaid the five biotypes that were identified from STRUCTURE (K = 5) on the haplotype network (Fig 2B). The result of haplotype analysis was highly discordant with the STRUCTURE results (see below). Among the five biotypes, red, blue, and green types were observed in both H2 and H9 haplotypes (Fig 2B). The majority of white type aphids belonged to H9, while blue and green types were mostly found in H2 (Fig 2B). Aphids with green type showed the most diverse haplotype diversity (Fig 2B). The haplotype H1 contained blue and dark blue types.

## Microsatellite data analysis

We successfully genotyped 578 aphid individuals of 36 HAPs of the two species using 9 microsatellite loci, and then found 463 non-clonal MLGs from all samples (Table 1). Generally, genetic diversity was high throughout the HAPs collected from woody perennials, which seemed to be regarded as the primary (overwintering) hosts. The mean number of alleles ($N_A$) and gene diversity ($H_S$) in *A. gossypii* host populations averaged (4.17 and 0.45), respectively, whereas *A. rhamnicola* populations averaged (4.98 and 0.48), respectively. Similarly, allelic richness ($R_S$) ($R_S$, mean ± s.d., 2.67 ± 0.67) in the *A. gossypii* populations was slightly lower than $R_S$ (2.79 ± 0.50) in those of *A. rhamnicola*. Surprisingly, among all HAPs, Ag_RH, Ag_HI, and Ar_CO had relatively very high $N_A$ at (8.56, 7.33, and 6.67), respectively, of which the $R_S$ values were also high at (4.62, 3.25, and 3.58), respectively. The expected heterozygosity ($H_E$) values in the *A. gossypii* populations ranged (0.21 to 0.70), whereas $H_E$ values in the *A. rhamnicola* populations ranged (0.30 to 0.59). In HWE, there were significant deviations in Ag_CA, Ag_CP, and Ar_PE by heterozygote excess, and in Ag_CJ, Ar_YO, Ar_PH, and Ar_RU by heterozygote deficit. Heterozygote excess in Ag_CA, Ag_CP, and Ar_PE were likely the result of heterosis or over-dominance related to selection preference toward heterozygous combination or fixation of heterozygous genotypes due to parthenogenesis of aphids in secondary host, especially under anholocyclic (permanently asexual) life [81]. Similar to our results, this phenomenon was already reported from several aphid species such as *Sitobion avenae*, *Myzus persicae* and *Rhopalosiphum padi* having permanently or temporary asexual life, which showed the significant heterozygote excess [82–84]. Negative *FIS* values also showed an increase in heterozygosity that was generally due to random mating or outbreeding, whereas positive *FIS* values explained that the amount of heterozygous offspring in the population decreased, usually due to inbreeding [85]. There was no evidence of significant linkage disequilibrium or frequency of null alleles.

**Genetic differentiation between host-associated populations and AMOVA.** We estimated pairwise genetic differentiation ($F_{ST}$) between 36 different HAPs of the two species (Table 2). The averaging pairwise $F_{ST}$ values among the HAPs of all, only *A. gossypii* (Ar_SE, Ar_PE, Ar_YO, and Ar_IX) and only *A. rhamnicola* were 0.329, 0.209 and 0.392, respectively. In *A. gossypii*, it appeared that the averaging pairwise $F_{ST}$ values among the different HAPs obtained from host plants within the same plant genus or family were relatively low, such as Cucurbitaceae (Ag_CU, Ag_CM; averaging pairwise $F_{ST}$ = 0.040), Solanaceae (Ag_SO, Ag_CA, Ag_CP; 0.029), *Euonymus* (Ag_EU, Ag_EJ; 0.016), and Asteraceae (Ag_ER, Ag_SN, Ag_CO; 0.025). Remarkably, Ag_PU (0.130), Ag_HI (0.152), Ag_FO (0.159), and Ag_RH (0.134), considered to be the primary host, showed relatively low average $F_{ST}$ values toward the other *A. gossypii* populations. In *A. rhamnicola*, Ar_LY, Ar_CB, and Ar_VE were genetically

**Table 2. Pairwise $F_{ST}$ divergence between 36 HAPs of the two species, *A. gossypii* groups (Ag) and *A. rhamnicola* groups (Ar).**

| | Ag_IL | Ag_CU | Ag_CM | Ag_KA | Ag_SO | Ag_CA | Ag_CP | Ag_PU | Ag_EL | Ag_HI | Ag_HR | Ag_EU | Ag_EJ | Ag_CI | Ag_FO | Ag_CE | Ag_ER | Ag_SN | Ag_CO |
|---|---|---|---|---|---|---|---|---|---|---|---|---|---|---|---|---|---|---|---|
| Ag_CU | 0.230 | | | | | | | | | | | | | | | | | | |
| Ag_CM | 0.183 | 0.040 | | | | | | | | | | | | | | | | | |
| Ag_KA | 0.202 | 0.145 | 0.092 | | | | | | | | | | | | | | | | |
| Ag_SO | 0.217 | 0.147 | 0.101 | 0.106 | | | | | | | | | | | | | | | |
| Ag_CA | 0.339 | 0.223 | 0.206 | 0.199 | 0.066 | | | | | | | | | | | | | | |
| Ag_CP | 0.352 | 0.209 | 0.187 | 0.163 | *0.049* | *-0.030* | | | | | | | | | | | | | |
| Ag_PU | 0.194 | 0.196 | 0.140 | 0.128 | 0.117 | 0.234 | 0.219 | | | | | | | | | | | | |
| Ag_EL | 0.387 | 0.267 | 0.203 | 0.140 | 0.224 | 0.337 | 0.311 | 0.216 | | | | | | | | | | | |
| Ag_HI | 0.169 | 0.137 | 0.060 | 0.090 | 0.121 | 0.244 | 0.216 | 0.111 | 0.196 | | | | | | | | | | |
| Ag_HR | 0.350 | 0.293 | 0.255 | 0.291 | 0.329 | 0.453 | 0.486 | 0.202 | 0.384 | 0.223 | | | | | | | | | |
| Ag_EU | 0.227 | 0.206 | 0.096 | 0.128 | 0.120 | 0.252 | 0.227 | 0.060 | 0.165 | 0.061 | 0.305 | | | | | | | | |
| Ag_EJ | 0.173 | 0.178 | 0.090 | 0.141 | 0.094 | 0.231 | 0.211 | 0.046 | 0.216 | 0.048 | 0.258 | 0.016 | | | | | | | |
| Ag_CI | 0.254 | 0.240 | 0.160 | 0.132 | 0.116 | 0.219 | 0.204 | 0.061 | 0.206 | 0.144 | 0.327 | 0.067 | 0.093 | | | | | | |
| Ag_FO | 0.234 | 0.260 | 0.159 | 0.184 | 0.149 | 0.300 | 0.295 | *0.017* | 0.266 | 0.106 | 0.235 | 0.064 | 0.040 | 0.087 | | | | | |
| Ag_CE | 0.299 | 0.295 | 0.194 | 0.169 | 0.215 | 0.332 | 0.312 | 0.134 | 0.114 | 0.152 | 0.283 | 0.090 | 0.124 | 0.119 | 0.137 | | | | |
| Ag_ER | 0.329 | 0.327 | 0.225 | 0.269 | 0.178 | 0.339 | 0.351 | 0.066 | 0.369 | 0.151 | 0.389 | 0.093 | 0.065 | 0.121 | 0.084 | 0.240 | | | |
| Ag_SN | 0.214 | 0.246 | 0.158 | 0.202 | 0.160 | 0.300 | 0.289 | 0.036 | 0.294 | 0.106 | 0.307 | *0.045* | 0.027 | 0.094 | 0.039 | 0.169 | *0.028* | | |
| Ag_CO | 0.204 | 0.240 | 0.161 | 0.179 | 0.146 | 0.287 | 0.274 | 0.015 | 0.279 | 0.100 | 0.237 | 0.057 | 0.033 | 0.086 | *-0.008* | 0.162 | 0.040 | *0.008* | |
| Ag_CL | 0.563 | 0.499 | 0.404 | 0.391 | 0.422 | 0.516 | 0.584 | 0.331 | 0.193 | 0.362 | 0.577 | 0.310 | 0.361 | 0.290 | 0.408 | 0.147 | 0.531 | 0.457 | 0.398 |
| Ag_CT | 0.432 | 0.353 | 0.279 | 0.311 | 0.271 | 0.346 | 0.386 | 0.191 | 0.371 | 0.257 | 0.435 | 0.214 | 0.209 | 0.176 | 0.254 | 0.241 | 0.297 | 0.232 | 0.232 |
| Ag_CJ | 0.234 | 0.255 | 0.205 | 0.225 | 0.201 | 0.308 | 0.301 | 0.119 | 0.342 | 0.170 | 0.312 | 0.176 | 0.127 | 0.213 | 0.136 | 0.264 | 0.213 | 0.126 | 0.115 |
| Ag_RH | 0.209 | 0.200 | 0.144 | 0.120 | 0.111 | 0.215 | 0.185 | 0.037 | 0.181 | 0.121 | 0.188 | 0.063 | 0.072 | 0.085 | 0.045 | 0.128 | 0.107 | 0.085 | 0.058 |
| Ar_SE | 0.473 | 0.439 | 0.400 | 0.351 | 0.344 | 0.417 | 0.403 | 0.304 | 0.411 | 0.361 | 0.515 | 0.359 | 0.335 | 0.356 | 0.367 | 0.379 | 0.435 | 0.392 | 0.365 |
| Ar_PE | 0.626 | 0.572 | 0.546 | 0.549 | 0.524 | 0.591 | 0.625 | 0.478 | 0.587 | 0.492 | 0.659 | 0.533 | 0.507 | 0.522 | 0.572 | 0.547 | 0.623 | 0.587 | 0.530 |
| Ar_YO | 0.469 | 0.411 | 0.411 | 0.397 | 0.421 | 0.479 | 0.468 | 0.374 | 0.466 | 0.387 | 0.496 | 0.412 | 0.395 | 0.428 | 0.448 | 0.457 | 0.491 | 0.435 | 0.415 |
| Ar_IX | 0.502 | 0.436 | 0.424 | 0.429 | 0.446 | 0.509 | 0.516 | 0.400 | 0.507 | 0.397 | 0.534 | 0.437 | 0.415 | 0.458 | 0.486 | 0.484 | 0.526 | 0.473 | 0.444 |
| Ar_RH | 0.463 | 0.428 | 0.383 | 0.338 | 0.348 | 0.437 | 0.393 | 0.347 | 0.396 | 0.330 | 0.490 | 0.337 | 0.352 | 0.382 | 0.387 | 0.390 | 0.449 | 0.403 | 0.383 |
| Ar_CO | 0.444 | 0.410 | 0.387 | 0.357 | 0.355 | 0.419 | 0.389 | 0.362 | 0.388 | 0.370 | 0.452 | 0.353 | 0.368 | 0.382 | 0.389 | 0.394 | 0.429 | 0.397 | 0.388 |
| Ar_LE | 0.474 | 0.437 | 0.413 | 0.368 | 0.411 | 0.491 | 0.443 | 0.376 | 0.398 | 0.348 | 0.509 | 0.363 | 0.369 | 0.410 | 0.430 | 0.394 | 0.481 | 0.427 | 0.410 |
| Ar_PH | 0.487 | 0.443 | 0.416 | 0.394 | 0.389 | 0.484 | 0.475 | 0.358 | 0.462 | 0.339 | 0.545 | 0.382 | 0.362 | 0.411 | 0.431 | 0.414 | 0.491 | 0.428 | 0.389 |
| Ar_ST | 0.356 | 0.301 | 0.191 | 0.239 | 0.313 | 0.444 | 0.458 | 0.263 | 0.367 | 0.095 | 0.440 | 0.216 | 0.208 | 0.295 | 0.294 | 0.274 | 0.387 | 0.304 | 0.270 |
| Ar_LY | 0.451 | 0.392 | 0.347 | 0.311 | 0.365 | 0.437 | 0.407 | 0.351 | 0.397 | 0.265 | 0.483 | 0.328 | 0.320 | 0.384 | 0.397 | 0.382 | 0.450 | 0.405 | 0.382 |
| Ar_CB | 0.406 | 0.361 | 0.321 | 0.282 | 0.355 | 0.425 | 0.395 | 0.344 | 0.366 | 0.239 | 0.436 | 0.318 | 0.315 | 0.363 | 0.377 | 0.355 | 0.432 | 0.389 | 0.367 |
| Ar_VE | 0.442 | 0.375 | 0.335 | 0.295 | 0.360 | 0.448 | 0.414 | 0.341 | 0.379 | 0.252 | 0.469 | 0.328 | 0.312 | 0.378 | 0.386 | 0.371 | 0.436 | 0.389 | 0.372 |
| Ar_RU | 0.451 | 0.417 | 0.383 | 0.381 | 0.389 | 0.463 | 0.442 | 0.352 | 0.421 | 0.333 | 0.456 | 0.343 | 0.336 | 0.381 | 0.380 | 0.376 | 0.417 | 0.388 | 0.362 |

| | Ag_CL | Ag_CT | Ag_CJ | Ag_RH | Ar_SE | Ar_PE | Ar_YO | Ar_IX | Ar_RH | Ar_CO | Ar_LE | Ar_PH | Ar_ST | Ar_LY | Ar_CB | Ar_VE |
|---|---|---|---|---|---|---|---|---|---|---|---|---|---|---|---|---|
| Ag_CT | 0.502 | | | | | | | | | | | | | | | |
| Ag_CJ | 0.520 | 0.340 | | | | | | | | | | | | | | |
| Ag_RH | 0.291 | 0.182 | 0.130 | | | | | | | | | | | | | |
| Ar_SE | 0.563 | 0.508 | 0.413 | 0.248 | | | | | | | | | | | | |
| Ar_PE | 0.720 | 0.667 | 0.561 | 0.399 | 0.549 | | | | | | | | | | | |
| Ar_YO | 0.617 | 0.521 | 0.407 | 0.307 | 0.494 | 0.570 | | | | | | | | | | |
| Ar_IX | 0.657 | 0.559 | 0.449 | 0.338 | 0.529 | 0.617 | 0.187 | | | | | | | | | |
| Ar_RH | 0.562 | 0.508 | 0.377 | 0.200 | 0.387 | 0.532 | 0.438 | 0.480 | | | | | | | | |
| Ar_CO | 0.492 | 0.456 | 0.381 | 0.209 | 0.383 | 0.440 | 0.418 | 0.447 | 0.089 | | | | | | | |
| Ar_LE | 0.560 | 0.531 | 0.407 | 0.294 | 0.410 | 0.428 | 0.454 | 0.470 | 0.292 | 0.317 | | | | | | |
| Ar_PH | 0.628 | 0.575 | 0.339 | 0.291 | 0.462 | 0.521 | 0.484 | 0.532 | 0.320 | 0.331 | 0.340 | | | | | |
| Ar_ST | 0.596 | 0.507 | 0.321 | 0.233 | 0.501 | 0.662 | 0.500 | 0.520 | 0.400 | 0.413 | 0.377 | 0.422 | | | | |
| Ar_LY | 0.573 | 0.495 | 0.403 | 0.267 | 0.418 | 0.570 | 0.455 | 0.463 | 0.295 | 0.327 | 0.253 | 0.384 | 0.245 | | | |
| Ar_CB | 0.509 | 0.467 | 0.391 | 0.280 | 0.423 | 0.535 | 0.446 | 0.441 | 0.314 | 0.347 | 0.253 | 0.360 | 0.178 | *0.000* | | |
| Ar_VE | 0.569 | 0.504 | 0.388 | 0.269 | 0.412 | 0.570 | 0.450 | 0.460 | 0.299 | 0.342 | 0.243 | 0.370 | 0.206 | *0.023* | 0.027 | |
| Ar_RU | 0.503 | 0.484 | 0.380 | 0.303 | 0.453 | 0.528 | 0.452 | 0.479 | 0.383 | 0.407 | 0.363 | 0.358 | 0.323 | 0.343 | 0.329 | 0.308 |

Values are significantly different from zero at $P < 0.05$ unless indicated as bold and italic.

close (0.050) to each other, despite belonging to the same host family/genus or being locally similar. In addition, Ar_RH was close to Ar_CO (0.089). Between *A. gossypii* and *A. rhamnicola* populations, Ar_ST and Ag_HI showed the lowest $F_{ST}$ value (0.095).

Two cases to confirm the genetic variance between the preordained groups were analyzed using AMOVA implemented in ARLEQUIN [68]. In the case of the analysis grouped by case 1, percentages of the genetic variance (PV) 'among groups' and 'among populations within groups' were 14.59% and 22.60%, respectively, which shows that there is some grouping effect by host plants, even though the majority of genetic variation was found 'among individuals within populations' as approximately 63% (Table 3). However, the genetic variance of about -1 ~ 0% 'among groups' in the both analyses grouped by cases 2 and 3 suggests that there are no grouped structures according to their lives in the perennial or non-perennial hosts on both *A. gossypii* and *A. rhamnicola* (Table 3). Interestingly, PV of 'among populations within groups' in *A. rhamnicola* was about 20% higher than that in *A. gossypii*, which means that the HAPs of *A. rhamnicola* is genetically differentiated further than those of *A. gossypii* (Table 3).

**Genetic similarity, structure, and assignment.** A plot of PCoA between 36 HAPs based on codominant genotypic genetic distances showed that the two species, *A. gossypii* and *A. rhamnicola*, were completely separated in each of the left, upper–right, and lower–right sides on the plot (Fig 3). Plots of *A. gossypii* populations being closely aggregated along the line of factor 1 means that they are genetically close to each other, whereas the plots of *A. rhamnicola* being relatively largely scattered show genetic isolations between them. Among all *A. gossypii* populations, Ag_HI and Ag_RH were relatively located near to the HAPs of *A. rhamnicola*, Ar_ST and Ar_SE, respectively. Plots of Ar_YO and Ar_IX, which had been taxonomically considered to *A. gossypii*, were closely located to each other, but distant from the majority group of *A. gossypii*.

The genetic structure of 36 HAPs of the two species (*A. gossypii* and *A. rhamnicola*) for 578 individuals was analyzed by STRUCTURE 2.3.3 [70]. In all STRUCTURE analyses from $K$ = (1 to 15), the most likely number of clusters was $K$ = 4, using the $\Delta K$ calculation according to the method of Evanno et al. [71]. Here, we show the structure results from $K$ = (2 to 5), in order to observe the change of genetic structure and assignment pattern according to the $K$ value (Fig 4). When $K$ = 2, the (first) white cluster dominantly appeared to *A. gossypii* populations, except for Ag_RH with a large green assignment, while the (second) green cluster was largely distributed among populations of *A. rhamnicola*. When $K$ = 3, the (first) white cluster also was dominant in *A. gossypii* HAPs, except for Ag_RH with large blue assignment and Ag_Hi with small green and blue ones, the (third) blue cluster as the '*Rhamnus* group' prevalent in Ar_SE, Ar_PE, Ar_YO, Ar_IX, Ar_RH, Ar_CO, Ar_LE, Ar_PH and Ar_ST, and the (second) green cluster in the rest as the '*Rubia* group'. When $K$ = 4, the genetic structure was basically similar

**Table 3. Analysis of molecular variance (AMOVA) results for microsatellite data analysis of aphids grouped by three cases: (1) *gossypii* vs *rhamnicola*, (2) perennial vs non-perennial host groups in *A. gossypii*, (3) perennial vs non-perennial host groups in *A. rhamnicola*.**

| Case | Among groups | | | Among populations within groups | | | Within populations | | |
|---|---|---|---|---|---|---|---|---|---|
| | $V_a$ | PV | *P* | $V_b$ | PV | *P* | $V_c$ | PV | *P* |
| 1 | 0.50 | 14.59 | <0.0001 | 0.77 | 22.60 | <0.0001 | 2.14 | 62.81 | <0.0001 |
| 2 | -0.01 | -0.55 | <0.0001 | 0.48 | 18.54 | <0.0001 | 2.13 | 82.01 | 0.4458 |
| 3 | -2.59 | -0.09 | 0.0001 | 1.47 | 41.55 | <0.0001 | 2.16 | 61.04 | 0.8602 |

$F_{st}$: Among groups, $F_{sc}$: Among populations within groups, $F_{ct}$: Within populations, V: Variance components. PV: Percentage of variation.

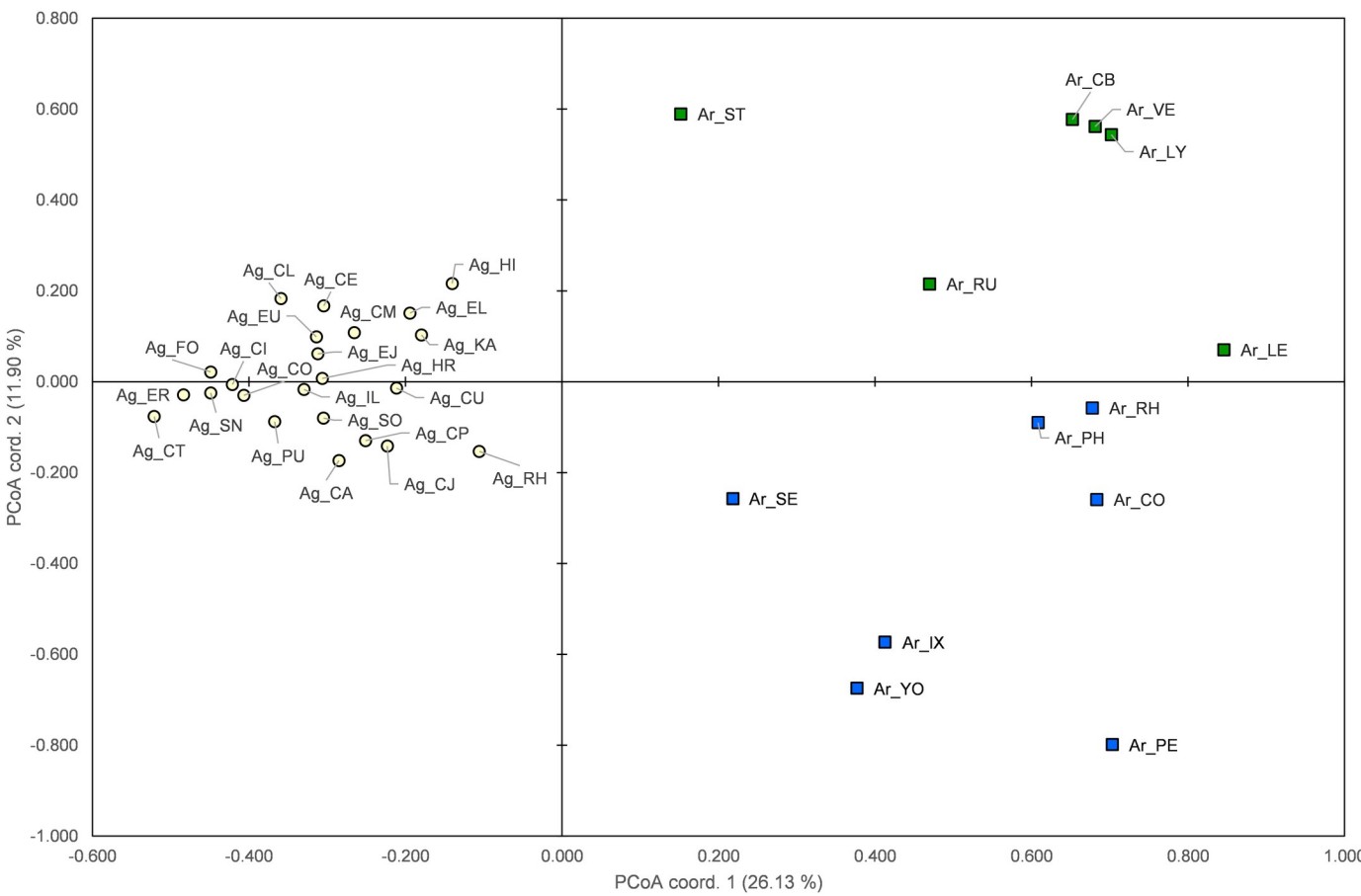

**Fig 3. A plot of the principal coordinate analysis based on the first two factors for 578 individuals of the four *gossypii* group species.** Each color corresponds to that shown in the results of STRUCTURE when *K* = 3 (Fig 3); white –23 HAPs of *A. gossypii*; blue–*Rhamnus* group, 7 HAPs of *A. rhamnicola*; green–*Rubia* group, 6 HAPs of *A. rhamnicola*. First and second coordinate axes account for (26.13 and 11.90) %, respectively.

to that at *K* = 3, except that the (fourth) red cluster was dominant in Ag_IL, Ag_CU, Ag_CM, Ag_KA, Ag_SO, Ag_CA, and Ag_CP, and partially appeared in Ag_EL, Ag_HI, Ag_HR, Ag_EU, Ag_EJ, and Ag_CI. When *K* = 5, the genetic structure was basically similar to that at *K* = 4, except that both Ar_YO and Ar_IX showed the (fifth) dark-blue cluster.

The Bayesian assignment tests using GENECLASS 2 [73] were carried out to identify the HAP (as population) membership of 578 individuals from all the 36 HAPs. The result of the assignment test (S2 Table) indicated the average probability with which individuals were assigned to the corresponding reference HAP (as population). The self-assignment probability values (SA) averaged (0.482 ± 0.106) (mean ± s.d.) in overall HAPs, (0.515 ± 0.103) in *A. gossypii*, and (0.427 ± 0.08) in *A. rhamnicola*. In *A. gossypii*, the mean assignment probability from 391 *A. gossypii* individuals into Ag_RH had the highest value (0.446, SA = 0.381), which was followed by the assignment value into each reference HAP of Ag_HI (0.219, SA = 0.478) and Ag_PU (0.214, SA = 0.458) (Fig 5). In *A. rhamnicola*, the mean assignment probability from 145 *A. rhamnicola* individuals into Ar_RU had the highest value (0.137, SA = 0.489), which was similar to the assignment rate into each reference HAP of Ar_CB (0.131, SA = 0.463) and Ag_LY (0.129, SA = 0.309) (Fig 6).

**Inferring a ancestral primary host to test hypothetical scenarios by ABC analysis.** To propose the most likely 'ancestral host evolution' scenario followed by the hypothesis that

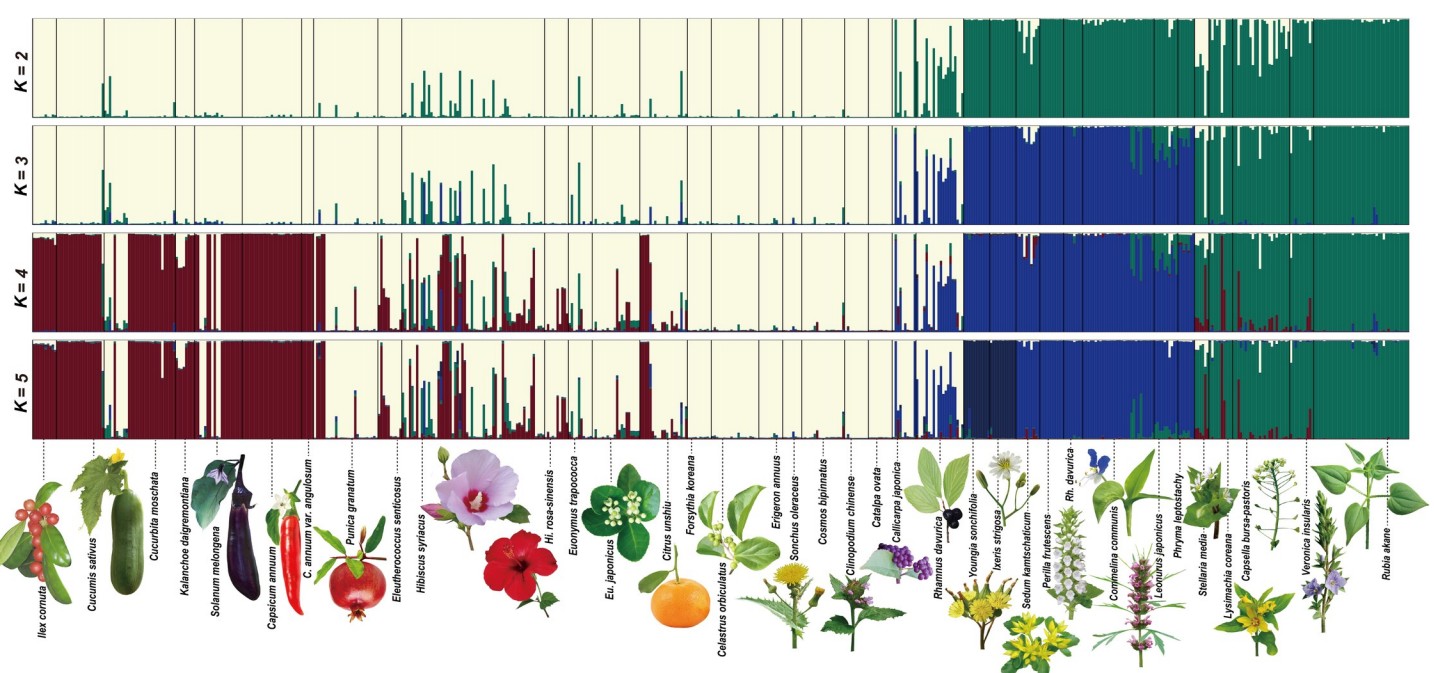

**Fig 4. Genetic structure of 36 HAPs of the two *gossypii* complex species (*A. gossypii* and *A. rhamnicola*) for 578 individuals performed by STRUCTURE 2.3.3 [70].** Results are shown for *K* = (2 to 5). Pop ID. (top) corresponds to Table 1, and the scientific plant name of each HAP is shown (bottom).

most of the *A. gossypii* populations originated from two possible ancestral HAPs (e.g. Ag_RH, Ag_HI), which had diverged from *A. rhamnicola*, the ABC test was conducted. We tested four scenarios to determine which HAP is the most ancestral among all the HAPs in *A. gossypii* (see "M&M"). The generated results are presented as a logistic regression using DIYABC software, estimating the PP of each tested evolutionary scenario of the hypothesis for the selected simulated data ($n_\delta$) (Cornuet et al. 2008), which ranged between (8 000 (or 6 000) and 80 000 (or 60 000)) $n_\delta$.

In the result of the first analysis (S4 Fig), scenario A1 obtained the highest PP ranging (0.664 ($n_\delta$ = 8000) to 0.697 ($n_\delta$ = 80 000)), with a 95% CI of (0.601–0.727) and (0.677–0.716).

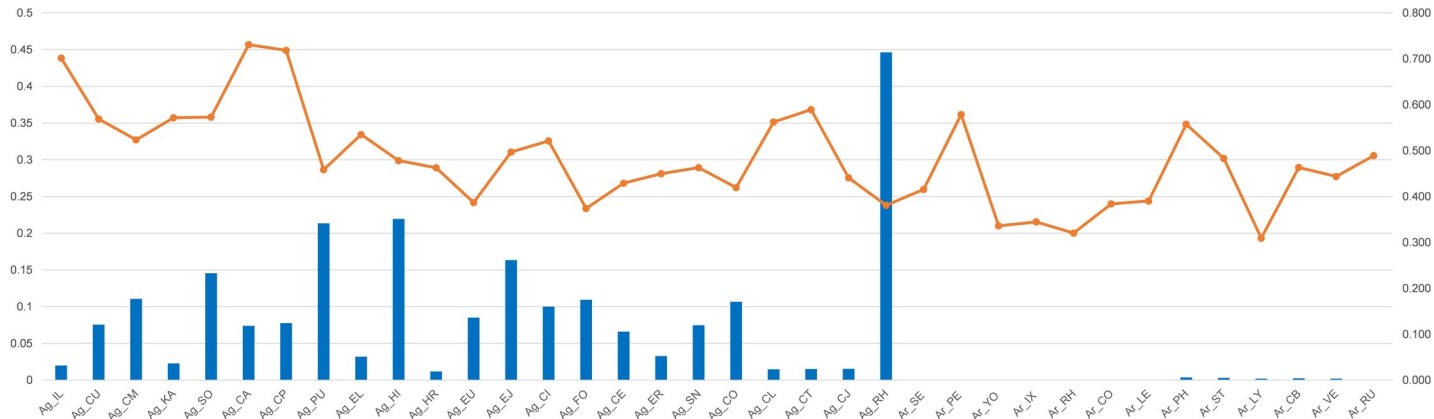

**Fig 5. Mean assignment rate (blue bar, values on left) from 391 *Aphis gossypii* individuals into each population (x column), and self-assignment rate (orange line, values on right) of individuals of each population using GENECLASS 2 [73].**

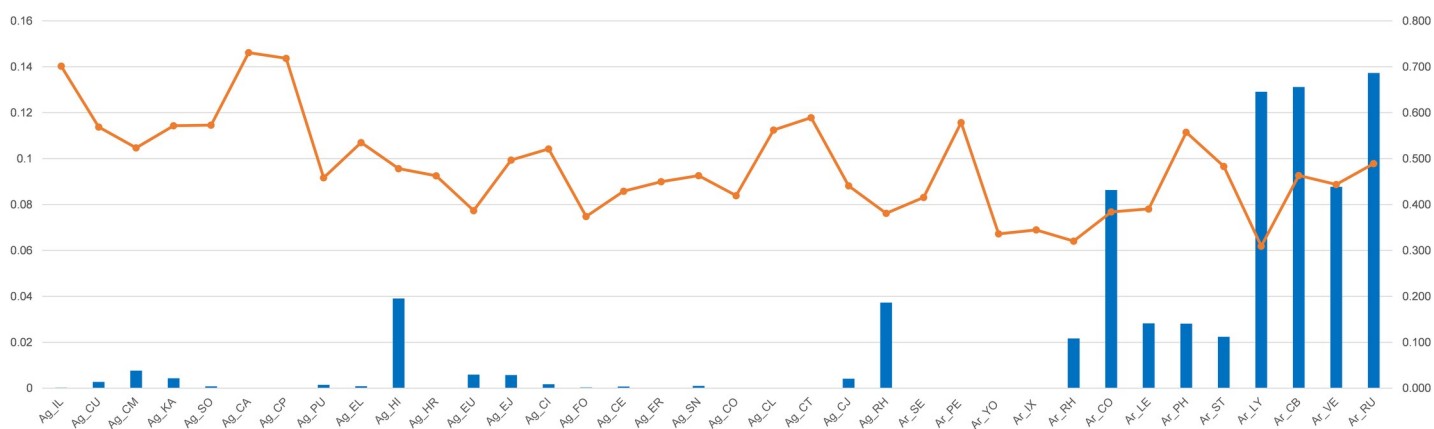

**Fig 6. Mean assignment rate (blue bar, values on left) from 187 *Aphis rhamnicola* individuals into each population (x column), and self-assignment rate (orange line, values on right) of individuals of each population using GENECLASS 2 [73].**

Scenario A2 showed a PP (Table 4). As a result, scenario A1 appeared as the most robust hypothesis with the highest PP among the four scenarios tested, which suggests that, compared to the other remaining hosts, *Rhamnus* is the most ancestral host for *A. gossypii* and *A. rhamnicola*, respectively.

In the result of the second analysis (S5 Fig), the scenario B2 was estimated more highly than the other four scenarios (Table 4). As a result, although the direct approach estimated a slightly higher PP for scenario B1 (0.520 and 0.480) than for B2 (0.460 and 0.448) (S5 Fig), the scenario B2 appeared as the the highest PP in the logistic regression. It is well supported that *A. rhamnicola* is the origin of *A. gossypii* (B1, B2), but is not conclusive whether the RED group in *A. gossypii* is diverged from the WHITE group, or vice versa.

In the result of the third analysis (S6 Fig), scenario C4 obtained the highest PP (Table 4). As a result, although the direct approach estimated a slightly higher PP for scenario C6 (0.400 and 0.326) and C5 (0.140 and 0.234) than for C4 (0.180 and 0.198) (S6 Fig), the scenario C4 appeared as the highest PP among the four scenarios tested in the logistic regression. This suggests that, within *A. gossypii*, the WHITE group is more ancestral than the MRW and RED groups, and then RED is originated from MRW, which hypothesizes that *Hibiscus* is the secondarily primary host, and can be still a refuge for the RED group.

**Table 4. Probabilities (with 95% confidence intervals in brackets) of the logistic regression for the scenarios in three different analyses inferred from DIYABC [77].**

| | Posterior probability of each historical scenario | | | | | |
|---|---|---|---|---|---|---|
| | First analysis (Scenario A#) | | Second analysis (Scenario B#) | | Third analysis (Scenario C#) | |
| No. | $n_\delta = 8000$ | $n_\delta = 80\,000$ | $n_\delta = 8000$ | $n_\delta = 80\,000$ | $n_\delta = 8000$ | $n_\delta = 80\,000$ |
| 1 | **0.6640 [0.6010,0.7269]** | **0.6965 [0.6767,0.7162]** | 0.4480 [0.3899,0.5061] | 0.4335 [0.4142,0.4529] | 0.0029 [0.0001,0.0056] | 0.0019 [0.0013,0.0024] |
| 2 | 0.0072 [0.0032,0.0112] | 0.0081 [0.0069,0.0094] | **0.5207 [0.4624,0.5790]** | **0.5169 [0.4974,0.5365]** | 0.0008 [0.0000,0.0017] | 0.0018 [0.0013,0.0024] |
| 3 | 0.3131 [0.2511,0.3751] | 0.2899 [0.2702,0.3095] | 0.0001 [0.0000,0.0002] | 0.0005 [0.0003,0.0006] | 0.2510 [0.1258,0.3762] | 0.2017 [0.1686,0.2349] |
| 4 | 0.0009 [0.0000,0.0023] | 0.0007 [0.0005,0.0009] | 0.0164 [0.0065,0.0263] | 0.0264 [0.0224,0.0303] | **0.3609 [0.2288,0.4929]** | **0.4917 [0.4460,0.5375]** |
| 5 | 0.0000 [0.0000,0.0001] | 0.0001 [0.0001,0.0001] | 0.0146 [0.0056,0.0236] | 0.0226 [0.0192,0.0260] | 0.1011 [0.0469,0.1554] | 0.0963 [0.0786,0.1139] |
| 6 | 0.0124 [0.0000,0.0259] | 0.0034 [0.0024,0.0044] | 0.0002 [0.0000,0.0007] | 0.0001 [0.0001,0.0001] | 0.2834 [0.1663,0.4004] | 0.2066 [0.1742,0.2389] |
| 7 | 0.0005 [0.0000,0.0012] | 0.0007 [0.0005,0.0009] | N/A | N/A | N/A | N/A |
| 8 | 0.0020 [0.0000,0.0042] | 0.0007 [0.0005,0.0008] | N/A | N/A | N/A | N/A |

For each comparison, the selected scenario (bold entry in shaded cell) was the one with the highest probability value.

## Discussion

### Complex evolution in *Aphis gossypii*

Our results identify the genetic structure between the various primary and secondary HAPs of the two species, *A. gossypii* and *A. rhamnicola*, encompassing the most various aphid samples from wild host plants. Our population genetic analyses reveal that *A. gossypii* and *A. rhamnicola* are mainly split into three (red, white, blue) and the other three (dark-blue, blue, green) biotypes, respectively, based on the STRUCTURE result (Fig 4, *K* = 5). The evolutionary trend of these aphids cannot be defined in any particular direction, and they show complex and various speciation tendencies. Here, we highlight major cases in these species.

One of the notable results is that some secondary HAPs seem to use a specific primary host (Fig 4, *K* = 4). In other words, *A. gossypii* and *A. rhamnicola* do not promiscuously use their primary and secondary host plants; instead, certain biotypes use only some secondary and specific primary hosts. For example, secondary HAPs having green biotype (e.g. *Capsella*, *Lysimachia*, *Stellaria*, and *Veronica*) seem to use only *Rubia* as the primary host in our dataset. On the other hand, *Rhamanus* serves as the primary host for the secondary HAPs having blue biotypes (e.g. *Commellina*, *Leonurus*, *Perilla*, *Phryma*, and *Sedum*). These cases indicate that a group that apparently uses several primary hosts is actually a complex of groups using a specific primary host.

In contrast to the previous cases, the white and red biotypes were found to share some primary and secondary hosts (Fig 4, *K* = 4). In particular, the white biotype has been extensively found in the most diverse primary hosts, such as *Callicarpa*, *Catalpa*, *Celastrus*, *Citrus*, *Euonymus*, *Hibiscus*, and *Punica*. The red biotype occurs in *Citrus*, *Eunonymus japonica*, *Hibiscus*, *Ilex*, and *Punica*. However, some primary hosts were exclusively occupied by the white (e.g. *Catalpa* and *Celastrus*,) or red type (e.g. *Ilex*), suggesting that these biotypes are possibly in a state of diverging through specialization to specific primary hosts. Interestingly, similar to the first case (i.e. blue and green types), the white and red types also tended to use specific secondary host groups, respectively. Except for a few secondary hosts (e.g. *Cucurbita* and *Solanum*), most of them represented only one biotype. For example, *Cucumis sativus* and *Capsicum annuum* were completely occupied by the red biotype. This is similar to the tendency found in most polyphagous aphids that the primary host is shared, but the secondary host is completely different [32,86–89].

In the STRUCTURE results, the dark-blue biotype (Fig 4, *K* = 5) represents the third case. The dark-blue biotype was represented only by two secondary hosts, *Ixeris* and *Youngia*, and was not found in any primary host. Thus, we assume that this case seems to be an ecologically isolated host race through the loss of a primary host. Although we did not confirm the lifecycle of this biotype in this study, there is a reference to *A. gossypii* inhabiting some Asteraceae plants in the previous study, even though those HAPs are identified to *A. rhamnicola* based on our results (Figs 3 and 4). Blackman and Eastop [22] found that populations producing eggs on the roots of *Ixeris*, including some Asteraceae plants in China identified as *A. gossypii*, may be other closely-related species. With large genetic differences from the main group of both *A. gossypii* and *A. rhamnicola* (Fig 4), they were possibly isolated to the secondary host directly from the ancestral primary HAP in *Rhamnus* by the host alternation, supporting the possibility of differentiation from their ancestral host race according to the loss of primary host [32]. Thus, the dark-blue biotype is likely to be an ecologically incipient species of *A. rhamnicola*, which has recently been derived by secondary host isolation.

Our results show strong evidence of ecological specialization through a primary host shift in both *A. gossypii* and *A. rhamnicola*. ABC analyses yielded the biotypes of the two species that were formed by shifting from the shared resource, *Rhamnus*, to different primary hosts,

respectively (S4 and S5 Figs). In particular, the series of primary host transitions identified in *A. gossypii* seem to have played an important role in the formation of their biotypes. For heteroecious aphids, a distinct choice of the primary host means not only utilizing different resources, but also genetic isolation between populations. This is because at one and the same time, the primary host is a resource, and a mating place. Accordingly, primary host selection in aphids is closely linked to genetic structure. Interestingly, in these species, primary host transitions occur more commonly than expected. As a traditional notion, primary hosts in aphids have been considered to be very fixed, and to not be able to easily escape, due to a highly adapted fundatrix morph [32,33]. In particular, the white biotype that appears in many *A. gossypii* uses a wide variety of taxonomically unrelated primary hosts, which show a variable relationship between primary and fundatrix (Fig 4). However, the results of our ABC analyses identified that *A. gossypii* was firstly derived from the biotype associated with *Rhamnus* to a white biotype, and then *Hibiscus* associated biotype was derived (S4 Fig). Thus, having multiple primary hosts is possibly a transitional step to shifting to another primary host. In fact, *Hibiscus* is a plant closely related to *Gossypium* (i.e. cotton), a representative secondary host of *A. gossypii*. Unfortunately, although *Gossypium* associated population was not included in this study, it can be inferred that there is a possibility that the transition of primary host through secondary host may have occurred. However, similar to our results, Carletto et al. [12] also suggested the possibility that *Hibiscus* was a shared ancestral host from which the agricultural divergence originated. In light of its HAP being genetically shared with the other HAPs in agricultural crops, such as *Gossypium*, cucurbits, and other secondary hosts.

Since the fundatrix specialization hypothesis [32,33] has been proposed, the complex lifecycle of aphids has long been regarded as a by-product of aphid evolution. However, the identification of several heteroecious HAPs in *A. gossypii* and *A. rhamnicola* in our study is largely in conflict with the expectations of this hypothesis. In our results, except for one case (i.e. the dark-blue biotype), the HAPs appear not to be genetically isolated completely but still to be linked together between some group of primary and secondary hosts, in contrast to the assumption that monoecy as a dead-end [32] is evolutionarily favorable over heteroecy. Moran's hypothesis [32] predicts that the dead-end of heteroecy always leads to specialization on the secondary host by loss of the primary host. Nevertheless, our results indicate that a new heteroecy race can commonly be derived from the heteroecy ancestors. In other words, our results show that lifecycle evolution is not a one-way process [32], but can be much more variable than we expected. These results are similar to the recent study on the genus *Brachycaudus* (Aphidinae: Macrosiphini), which provided strong evidence of the evolutionary lability of a complex lifecycle in *Brachycaudus* [89]. In addition, the use of several primary hosts found in some races (i.e. red and white biotypes) negates the core assumption of the fundatrix specialization hypothesis [32,33] that the fundatrix is fully adapted to the only primary host, and is inadequate to other hosts. Using multiple primary hosts is possibly a strategy for their migration success. Indeed, a migration failure can lead to high risk. For example, aphids using only a single primary host, such as *Rhopalosiphum padi*, have only a 0.6% migration success rate [90].

### Ancestral host association in *A. gossypii* complex

*Rhamnus* appears to be the most ancestral host plant for both *A. gossypii* and *A. rhamnicola*. Several species in the *gossypii* complex group have intimate relationships with *Rhamnus* (e.g. *A. frangulae*, *A. glycines*, *A. gossypii*, *A. nasturtii* and *A. rhamnicola*), which have been considered *Rhamnus* as an ancestral host for this aphid group [22,43]. Our ABC result is consistent with this assumption (S4 and S5 Figs). As in the previous study, *Rhamnus* appears to serve as a shared primary host for both *A. gossypii* and *A. rhamnicola*. In the GENECLASS2 analyses

(Figs 5 and 6) as well, the assignment of most of the *gossypii* HAPs to *Rhamnus* was very high, corroborating that it was differentiated from the ancestral host, *Rhamnus*.

Despite the differentiation of aphids into various species using the different hosts (mainly secondary hosts), host utilization of *Rhamnus* still remains in several species of the *gossypii* group. The phylogenetic studies of *Aphis* showed that heteroecious species using *Rhamnus* as the primary host were derived non-consecutively from monoecious species [44,91]. In other words, even if a monoecious species has been derived by loss of heteroecy, it seems likely to not be the dead-end of evolution [33], as well as a complete disconnect from the ancestral primary host. For example, *A. rhamnicola* and *A. gossypii* are heteroecious species, which use *Rhamnus* as the primary host, and several monoecious species on various host plants appear to have been derived between them [46]. Our ABC analyses confirmed that *Rhamnus* was lost once when branching from blue type to green type, and was then regained in white type (S4 Fig). Surprisingly, these ABC results, similarly supported by the GenClass2 results (Figs 5 and 6), almost coincided with our haplotype network results (Fig 2).

These results conflict the fundatrix specialization hypothesis [32,33], which predicts that once aphids leave the ancestral primary host, they cannot regain it again. Recently, the phylogenetic study of *Brachycaudus* demonstrated that even if they lost their potential ancestral Rosaceae hosts, they can easily regain their hosts to be the primary host for heteroecy, or the sole host for monoecy [89]. The ancestral primary host does not seem to be an absolute being that cannot be changed due to the adaptation of the fundatrix, but seems to be a conserved resource within a specific aphid group. In fact, such a labile of aphid lifecycle related to the use of primary hosts may also occur within a species. Host alternation for some species is often not obligatory but facultative, in which the migration to the secondary host can often be omitted [15,92]. As an example, a facultative alternation lifecycle has been reported in populations of *Aphis fabae*, even although the vast majoriy of them migrate routinely between primary and secondary hosts [92]. Although there is little known about the facultative use of the primary host in *A. gossypii*, it may be related to the primary host range expansion and lifecycle lability.

## The evidence of hybridization between *A. gossypii* and *A. rhamnicola*

Our population genetic analyses based on microsatellite and *COI* gene show that there is a significant conflict between the two results. Regardless of the primary and secondary hosts, we found individuals that are difficult to identify in some host-associated populations. *A. gossypii* and *A. rhamnicola* appeared to share major haplotypes, H9 and H2, respectively, of their counterpart species with each other. Although the PCoA and STRUCTURE results (Figs 3 and 4) based on the microsatellites clearly showed identification of *A. gossypii*, the individuals corresponding to H2 (major haplotype of *A. rhamnicola*) were two individuals from Ag_Hi and one from Ag_KA, whereas the individuals corresponding to H9 (major haplotype of *A. gossypii*) were also identified as *A. rhamnicola*, but six from Ar_CB, one from Ar_PH, and four from Ar_RU. Surprisingly, the cross-sharing haplotypes (H5, H9) between these two species unexpectedly contained several kinds of both primary and secondary hosts.

Comparing the host races between *A. gossypii* and *A. rhamnicola*, most of them were distributed in two haplotypes (H9 and H2), and they were clearly identified as distinct species, based on the microsatellite analysis (Fig 2). However, a number of intermediate haplotypes, H13, H11, H12, H10, and H18, were observed among the species (Fig 2). The H1 haplotype shared by several wild plant populations, such as *Ixeris*, *Leonurus*, *Perilla*, *Phryma*, and *Youngia*, is closely related to the H9 (major haplotype of *A. gossypii*). However, according to our microsatellite data, these populations appear to be closer to *A. rhamnicola* (Figs 3 and 4). Similarly, collected from *Rubia akane*, individuals of H4, H6, H7, and H8 haplotypes

apparently have alleles of *A. rhamnicola* in microsatellite data; it is most unusual that they have the haplotypes closely related to and derived from *A. gossypii*, rather than *A. rhamnicola*. In the case of H18, it is inferred to be the similar haplotype of true *A. gossypi* in Curcubitaceae, and H13 of Ar_SE, which was often cryptically recognized from *A. gossypii* as *A. rhamnicola* based on morphology and mtDNA [42,93], was nested between the two species, *A. gossypii* and *A. rhamnicola*. Since it was reported that populations with a sexual phase on *Rubia cordifolia* in Japan appeared to be isolated from those on other primary hosts [25], it is suspected that most of the populations collected from *Rubia* in the past might actually be host races of *A. rhamnicola*, but misidentified.

Hybridization can have important evolutionary consequences, including speciation in association with novel host plants in insects [94]. In our study, as the two species, *A. gossypii* and *A. rhamnicola*, with distinct taxonomic and phylogenetic differences shared *COI* haplotypes of counterpart species with each other, there is a possibility of introgression by hybridization between them. These two species share overwintering (primary) host, such as *Rhamnus*, so mating and reproducing contemporarily in the same leaves or branches, because there are no physiological or ecological significant differences [43,46]; there is therefore the possibility that hybridization occurs between them. In fact, interspecific cross mating between sexuparae of *A. glycines*, *A. gossypii*, and *A. rhamnicola* in *Rhamnus* spp. has often been observed (unpublished data). It is an interesting phenomenon that a hybrid zone mediated not by geography, but by a resource, can exist. Lozier et al. [87] first detected hybridization and introgression between plum and almond associated *Hyalopterus* spp. on these host plants, which surprisingly were capable of feeding and developing on apricot from each species. For that possibility of hybridization, it was suggested that imperfections in any number of mechanisms associated with host plant choice [95,96] could lead to strong selection against hybrids on parental host plants, but less so on apricot [87]. Although apricot was introduced later than other host species in the studied area, it remains a mystery why only it is able to attract all *Hyalopterus* groups and permit hybridization, whereas the other *Prunus* hosts are more restrictive [87]. Based on the phylogenetic results of *COI* or *Buchnera 16S* for *Hyalopterus* spp. [87], peach or apricot could be inferred to their ancestral host like the *Rhamnus* as a hybridization host utilized by the *gossypii* group. These results from the *gossypii* group reaffirm the hypothesis of Lozier et al. [87], which corroborates that such hybridization in the aphid group often occurs by co-existence in the primary host. However, further research is needed to determine whether a primary host (i.e. hosts that can be utilized by various host races, where they co-exist and overwinter together) are ancestral or derivational for those aphids.

## Supporting information

**S1 Fig.** The first eight scenarios (A1–A8) for the DIYABC analyses to infer the host evolution of the two *Aphis* species, using a dataset that includes 578 individuals from four population groups, which consisted of 75 individuals from the 'BLUE' group (Ar_SE, Ar_PE, An_IX, An_YO, Ar_CO, Ar_PH, Ar_RH, Ar_LE); 90 from the 'GREEN' group (Ar_ST, Ar_VE, Ar_LY, Ar_CB, Ar_RU); 30 from the 'MIXBW (BLUE+WHITE)' group (Ag_RH, Ag_CJ); and 361 from the 'WHITE' group (Ag-IL, Ag_CE, Ag_EU, Ag_EJ, Ag_PU, Ag_CU, Ag_CM, Ag_KA, Ag_EL, Ag_HI, Ag_HR, Ag_FO, Ag_CI, Ag_ER, Ag_SN, Ag_CO, Ag_SO, Ag_CA, Ag_CP, Ag_CL, Ag_CT).
(TIF)

**S2 Fig.** The second six scenarios (B1–B6) for the DIYABC analyses to infer the host evolution of the two *Aphis* species, using a dataset that includes 311 individuals from four population groups, which consisted of 75 individuals from the 'BLUE' group (Ar_CO, Ar_PH, Ar_RH,

Ar_SE, Ar_PE, Ar_LE); 90 from the 'GREEN' group (Ar_ST, Ar_VE, Ar_LY, Ar_CB, Ar_RU); 60 from the 'RED' group (Ag_IL, Ag_CU, Ag_CA, Ag_CP); and 86 from the 'WHITE' group (Ag_CE, Ag_FO, Ag_ER, Ag_SN, Ag_CO, Ag_CL, Ag_CT).
(TIF)

**S3 Fig.** The third six scenarios (C1–C6) for the DIYABC analyses to infer the host evolution of *Aphis gossypii*, using a dataset that includes 391 individuals from four population groups except for BLUE and GREEN groups in the first and second analysis, which consisted of 30 individuals from the 'MBW (BLUE+WHITE)' group (Ag_RH, Ag_CJ); 207 from the 'MRW (RED+WHITE)' group (Ag_EU, Ag_EJ, Ag_PU, Ag_SO, Ag_CM, Ag_EL, Ag_HI, Ag_HR, Ag_CI); 68 from the 'RED' group (Ag-IL, Ag_CU, Ag_KA, Ag_CA, Ag_CP); and 86 from the 'WHITE' group (Ag_CE, Ag_FO, Ag_ER, Ag_SN, Ag_CO, Ag_CL, Ag_CT).
(TIF)

**S4 Fig.** Plots output by DIYABC showing the PP (y-axis) of the first eight scenarios (A1–A8) through the direct estimate (left), and the logistic regression (right) approaches, as output by DIYABC. The x-axis corresponds to the different $n_\delta$ values used in the computations. The results have been obtained by performing the first analysis with four scenarios.
(TIF)

**S5 Fig.** Plots output by DIYABC showing the PP (y-axis) of the second six scenarios (B1–B6) through the direct estimate (left), and the logistic regression (right) approaches, as output by DIYABC. The x-axis corresponds to the different $n_\delta$ values used in the computations. The results have been obtained by performing the first analysis with four scenarios.
(TIF)

**S6 Fig.** Plots output by DIYABC showing the PP (y-axis) of the third six scenarios (C1–C6) through the direct estimate (left), and the logistic regression (right) approaches, as output by DIYABC. The x-axis corresponds to the different $n_\delta$ values used in the computations. The results have been obtained by performing the first analysis with four scenarios.
(TIF)

**S1 Table. Collection data for 578 aphids analyzed in this study.** [†] possibly *A. rhamnicola* or other cryptic species [††] possibly other cryptic species.
(DOCX)

**S2 Table. Mean assignment rate of individuals into (rows) and from (columns) each population using GeneClass 2 [73].** Values in bold indicate the proportions of individuals assigned to the source population (self-assignment). Values less than 0.001 were excluded from the table.
(DOCX)

**S1 File.**
(ZIP)

## Author Contributions

**Conceptualization:** Yerim Lee, Thomas Thieme, Hyojoong Kim.

**Data curation:** Yerim Lee, Hyojoong Kim.

**Formal analysis:** Yerim Lee, Hyojoong Kim.

**Funding acquisition:** Hyojoong Kim.

**Investigation:** Thomas Thieme, Hyojoong Kim.

**Methodology:** Yerim Lee, Hyojoong Kim.

**Project administration:** Hyojoong Kim.

**Resources:** Yerim Lee, Thomas Thieme, Hyojoong Kim.

**Software:** Yerim Lee, Hyojoong Kim.

**Supervision:** Hyojoong Kim.

**Validation:** Yerim Lee, Hyojoong Kim.

**Visualization:** Yerim Lee, Hyojoong Kim.

**Writing – original draft:** Yerim Lee, Thomas Thieme, Hyojoong Kim.

**Writing – review & editing:** Yerim Lee, Thomas Thieme, Hyojoong Kim.

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
