## [Decision Letter · Decision Letter 0]

28 Aug 2020

PONE-D-20-16606

Complex evolution in Aphis gossypii group (Hemiptera: Aphididae), evidence of primary host shift and hybridization between sympatric species

PLOS ONE

Dear Dr. Kim,

Thank you for submitting your manuscript to PLOS ONE. After careful consideration, we feel that it has merit but does not fully meet PLOS ONE’s publication criteria as it currently stands. Therefore, we invite you to submit a revised version of the manuscript that addresses the points raised during the review process.

Both reviewers have pointed out that the fundamental conclusions of this paper regarding the evolution of are contingent on the accuracy of the input data, whether it be the definition of primary hosts (Reviewer 1), confirmation of sexual reproduction (Reviewer 1), and any uncertainty in the phylogeny used in the analyses (Reviewer 2).  The authors should address any ambiguities in their conclusions arising from uncertainties in this input data.Reviewer 1 has raised extensive concerns about the (over)-interpretation of the population genetic data.  The authors must address these comments, and should again make it clear in the Discussion whether any ambiguities or uncertainty arise in their conclusions as a consequence.The authors should also consider the recommendations of Reviewer 1 to improve the description of sections of the population genetic analyses to make it more comprehensible to a non-specialist audience.

We look forward to receiving your revised manuscript.

Kind regards,

Owain Rhys Edwards, Ph.D.

Academic Editor

PLOS ONE

Journal Requirements:

"NO authors have competing interests."

We note that one or more of the authors are employed by a commercial company: BTL Bio-Test Labor GmbH.

2.1. Please provide an amended Funding Statement declaring this commercial affiliation, as well as a statement regarding the Role of Funders in your study. If the funding organization did not play a role in the study design, data collection and analysis, decision to publish, or preparation of the manuscript and only provided financial support in the form of authors' salaries and/or research materials, please review your statements relating to the author contributions, and ensure you have specifically and accurately indicated the role(s) that these authors had in your study. You can update author roles in the Author Contributions section of the online submission form.

2.2. Please also provide an updated Competing Interests Statement declaring this commercial affiliation along with any other relevant declarations relating to employment, consultancy, patents, products in development, or marketed products, etc.  

Reviewers' comments:

Reviewer's Responses to Questions

**Comments to the Author**

1. Is the manuscript technically sound, and do the data support the conclusions?

Reviewer #1: Partly

Reviewer #2: Partly

2. Has the statistical analysis been performed appropriately and rigorously? 

Reviewer #1: I Don't Know

Reviewer #2: I Don't Know

3. Have the authors made all data underlying the findings in their manuscript fully available?

Reviewer #1: No

Reviewer #2: Yes

4. Is the manuscript presented in an intelligible fashion and written in standard English?

Reviewer #1: No

Reviewer #2: Yes

5. Review Comments to the Author

Reviewer #1: Review of PONE-D-20-16606

This study investigates the complex situation of different host races/subspecies/species within the A. gossypii group with over 500 individuals of A. gossypii and A. rhamnicola collected from 36 different plants, mainly in Korea. Mitochondrial haplotyping (COI, barcoding region) is combined with microsatellite genotyping to better understand their relationships. Of particular interest is the evolution of life-cycles, for example if host-alternating (heteroecious) taxa always give rise to monoecious taxa through the loss of the primary host, or if primary hosts can also be re-gained or changed over evolutionary time. Using Approximate Bayesian Computing to compare the likelihood of different scenarios, the authors also try to identify the ancestral primary host of the complex. The analyses confirm previous work in identifying Rhamnus as the ancestral primary host, but they indicate that counter to common belief, aphid life-cycles are quite labile. There is not only unidirectional evolution from heteroecy to monoecy. The author’s scenarios suggest that heteroecious races can derive from other heteroecious races through a shift in the primary hosts, or that primary hosts can be re-gained.

Although I find the main results interesting, I have some concerns with this study. A lot of the results hinge on the correct assignment of plant species as primary or secondary hosts of the aphids. It is completely unclear how this was ascertained. The primary host is defined as the plant on which the aphids mate and lay their diapausing eggs. Did the authors really verify that aphids from plants identified as primary hosts were indeed reproducing sexually there, or was the timing of the sampling at least such that this could be safely assumed (late fall/early spring, just before egg laying or after hatching). It is quite common to sometimes find aphids on some woody hosts that are not necessarily their primary hosts. Misassignment of individuals to ‘their’ primary host could clearly lead to different interpretations. Please elaborate how this was excluded.

Completely lacking is a discussion of the relationship between the loss of sex and the loss of the primary host from the aphid life-cycle. Quite a few host-alternating aphids omit the sexual generation of their life-cycle in regions with mild climates that permit parthenogenetic overwintering or where primary host plants are not available. This will not necessarily lead to the evolution of a new host race, at least not immediately. A lot of the pest populations of A. gossypii worldwide consist of just a few permanently asexual clones with strong host associations. There were a number of studies on those by a French team led by Vanlerberghe-Masutti & colleagues, a literature that should be better integrated in this paper.

My main concern, however, is the presentation of the population genetic analyses and partially their (over-) interpretation. The Results section is extremely hard to follow and requires major changes to make it more accessible to an average reader. Some English language editing would also help with that.

My issues start with the standard population genetic analyses of the microsatellite data and their interpretation. First of all, everything on lines 433-441 is completely speculative without supporting evidence. There is simply no way of telling just from the patterns whether significant deviations from HWE in some subpopulations are due heterosis or any other mechanism. Some other statements are plain wrong, e.g. that “an increase in heterozygosity that was generally due to random mating or outbreeding”. Random mating is what restores HWE in a population! I think the authors would better restrict themselves to the description of the patterns. Secondly, some deviations from HWE may simply be due to the inclusion of multiple copies of the same genotype (clone). Generally, clonal diversity is high in these samples, but in Ag_CA, for example, there are just six different MLGs among 25 individuals. This sample cannot be in HWE for purely statistical reasons. If just one relatively heterozygous genotype occurs multiple times in this sample, there is likely to be a significant heterozygote excess. I would thus recommend to test for deviations from HWE also with a dataset reducing clonal copies, i.e. with only one representative of each MLG per sample.

Then I find the verbal account of the pairwise genetic differentiation (Fst) results very hard to follow (l. 444-457). The second sentence, for example, makes no sense to me. Why pick out four particular populations for “the HAPs of A. gossypii”, calculate some average Fst between them and A. rhamnicola (which populations, all of them?), and then only come up with three values? I really cannot follow. The whole business of somehow averaging pairwise Fst values is very confusing. Please re-structure this whole passage. Maybe you can get by by describing the main patterns from Table 2 rather than work with some difficult to trace averages.

Similarly confusing is the passage reporting the AMOVA results (l. 458 – 465). The groupings are not properly explained, neither in the Methods nor here. The number of df in the AMOVA table suggests there were four groups for ‘host plant’, but what were these? The same four picked in the paragraph above (for unknown reasons), or was it plant genera/families (Cucurbitaceae, Solanaceae, Euonymus, Asteraceae)? Please clarify.

In the passage reporting the results of the assignment tests with GENECLASS, it is unclear what the first values in the brackets before the self-assignment probabilities (SA) represent and why they are relevant. Please explain.

Finally, the results text on the ABC analysis comparing different evolutionary scenarios is very hard to follow. There are literally two full pages of sentences like these: “Scenario A3 showed a PP ranging (0.313 (nδ = 8 000) to 0.290 (nδ = 80 000)), with a 95 % CI of (0.251–0.375) and (0.270–0.309). Scenario A4 showed a PP ranging (0.001 (nδ = 8 000) to 0.001 (nδ = 80 000)), with a 95 % CI of (0.000–0.002) and (0.001–0.001).” With the corresponding figures all hidden in the electronic appendix, this is all but unreadable. The results are interesting, so my suggestion would be to maybe present the analysis results in the form of a table, and combine this with a figure at least of the best-supported evolutionary scenario in the paper, not the appendix (like the different plots in Fig S1). This would make the results more accessible. A clearer explanation of the nδ mumbers is also required (number of simulated datasets considered). Why look at two different numbers, and why do these numbers not correspond to those mentioned in the Methods section?

Issues of over-interpretation also extend to the discussion, for example “they were clearly identified as disctinct species, based on the microsatellite analysis”. There is not really an established straightforward way inferring species status just from genetic differentiation at microsatellite loci.

Although the authors declare that all data will be publicly available, I could not find any statement in the PDF about where the data are or will be made accessible.

Minor comments:

l. 26: tested to confirm -> used to infer

l. 27: most primitive -> ancestral

l. 28: delete ‘respectively’

l. 31 (and elsewhere): heteroecy (noun) -> heteroecious (adjective)

l. 34: delete ‘of counterpart species’

l. 48: Jaenike 1990 in AnnuRevEcolSyst also seems like a key reference here.

l.85-87: Unclear sentence. Please re-word.

l. 110: what does ‘primitive’ mean in this context?

l. 111 and elsewhere: adaptive -> adapted

l. 131: Again, I think ancestral would be more appropriate than primitive.

Table 2: Host-race populations may be an undue inference. Maybe just call them ‘host-associated’?

Fig. 3: If color-coding points with reference to the STRUCTURE plot for K=3 in Fig. 4, why not use the same colors for all groups?

l. 514-515: One of these K should be something other then 5, right?

l. 547: likelihood -> likely

l. 720-721: Just like this the statement is incorrect (does not often migrate…). Aphis fabae is also a complex of subspecies with a shared primary host and different secondary host ranges, but the vast majority of them does migrate routinely between primary and secondary host, at least in climates with a cold winter.

Reviewer #2: This study focusses on two species of aphids, A. gossypii and A. rhamnicola. Both are members of a confusing species complex, the frangulae group, many of which use Rhamnus as a primary host. The authors newly sampled many populations of both species from a much broader range of host plants than has been done before. They conducted various population genetic analyses from mitochondrial COI barcode sequences and from multiple microsatellite loci. One goal was to determine whether populations might fall into discrete genetic entities that use specific host plants (or specific sets of host plants). Another goal was to identify the pattern of shifts between host plants and implications for life cycle evolution, e.g., whether heteroecious life cycles could be derived directly from other heteroecious life cycles on a different primary host. A final goal was to infer the ancestral primary host plant.

I think a major contribution of this work is in the identification of the host-associated populations—that is, that both of these species sort out into biotypes that are fairly specific to certain host plants. They are not each a randomly, highly polyphagous entity.

Except for the COI haplotype network and principal coordinate methods, I was unfamiliar with the data analysis for microsatellites, so I can’t comment on the specifics of those, other than one point (below). The haplotype and PCoA methods seemed fine.

One point in the ABC methods and analysis that concerns me is setting A. rhamnicola in the ancestral position of the genealogy (lines 323-325). This was based on previous findings of reference #41 (Lee Y, Lee W, Lee S, Kim H. A cryptic species of Aphis gossypii (Hemiptera: Aphididae) complex revealed by genetic divergence and different host plant association. Bull Entomol Res. 2015;105(1):40-51). However, the tree in that paper is an unrooted neighbor-joining distance dendrogram, not a true character-based rooted phylogeny. That aside, what is more pertinent is that A. rhamnicola is not located in a basal position in that tree, but is nested in a more derived position—but actually since the tree is technically unrooted we don’t really know where A. rhamnicola is placed relative to the root. Furthermore, nearly all of the inter-species relationships in the tree are unsupported, with bootstrap values below 60-50%. If the ABC results are dependent on which entities are designated as “ancestral” then the findings from these tests could be in question. I urge the authors to acknowledge the uncertainty in relationships in this species group and determine if it affects their results.

In the Discussion, the authors interpret the cross-sharing of haplotypes and microsat alleles as hybridization. However, given that the relationships between gossypii, rhamnicola, and related species are uncertain (see above), it seems that another explanation for shared haplotypes and alleles might be incomplete lineage sorting. The authors should consider this (and discount if they can).

Minor points:

• avoid the term “primitive”. Use “ancestral” instead.

• Figure 2B caption needs correcting (it repeats 2A caption)

• Is the microsatellite raw data deposit somewhere?

6. PLOS authors have the option to publish the peer review history of their article (what does this mean?). If published, this will include your full peer review and any attached files.

Reviewer #1: No

Reviewer #2: No

---

## [Author Response · Author response to Decision Letter 0]

23 Oct 2020

E1. Please ensure that your manuscript meets PLOS ONE's style requirements, including those for file naming. The PLOS ONE style templates can be found at

>[E#1 Response]

I double-checked ‘the PLOS ONE style templates’ and confirmed that there is no problem. All file names have been changed according to the regulations. Thank you.

E2. Thank you for stating the following in the Competing Interests section: "NO authors have competing interests." We note that one or more of the authors are employed by a commercial company: BTL Bio-Test Labor GmbH.

 2.1. Please provide an amended Funding Statement declaring this commercial affiliation, as well as a statement regarding the Role of Funders in your study. If the funding organization did not play a role in the study design, data collection and analysis, decision to publish, or preparation of the manuscript and only provided financial support in the form of authors' salaries and/or research materials, please review your statements relating to the author contributions, and ensure you have specifically and accurately indicated the role(s) that these authors had in your study. You can update author roles in the Author Contributions section of the online submission form. 

Please also include the following statement within your amended Funding Statement. “The funder provided support in the form of salaries for author [insert relevant initials], but did not have any additional role in the study design, data collection and analysis, decision to publish, or preparation of the manuscript. The specific roles of these authors are articulated in the ‘author contributions’ section.” If your commercial affiliation did play a role in your study, please state and explain this role within your updated Funding Statement.

>[E#2 Response]

Although Thomas Thieme is employed by a commercial company: BTL Bio-Test Labor GmbH, This research has nothing to do with the company he is employed by. As you mentioned, the company did not play a role in the study design, data collection and analysis, decision to publish, or preparation of the manuscript and only provided financial support in the form of TT's salaries. Thus, we insert that statement as below. “The funder provided support in the form of salaries for Thomas Thieme, but did not have any additional role in the study design, data collection and analysis, decision to publish, or preparation of the manuscript. The specific roles of these authors are articulated in the ‘author contributions’ section.”

E3. In your Methods section, please provide additional information regarding the permits you obtained for the work. Please ensure you have included the full name of the authority that approved the field site access and, if no permits were required, a brief statement explaining why.

>[E#3 Response]

We insert the statement as below: “As all collections have not been carried out in restricted areas, national parks, etc. where permits are required, it is clearly stated that there is no content regarding collection permits.” (line 165)

E4. We note that you have included the phrase “data not shown” in your manuscript. Unfortunately, this does not meet our data sharing requirements. PLOS does not permit references to inaccessible data. We require that authors provide all relevant data within the paper, Supporting Information files, or in an acceptable, public repository. Please add a citation to support this phrase or upload the data that corresponds with these findings to a stable repository (such as Figshare or Dryad) and provide and URLs, DOIs, or accession numbers that may be used to access these data. Or, if the data are not a core part of the research being presented in your study, we ask that you remove the phrase that refers to these data.

>[E#4 Response]

“Data not shown” appears once in the manuscript as follows. [P12 line 223. In the preliminary study, we had already checked the cross-species amplification test of these loci on A. fabae, Hyalopterus pruni, Rhopalosiphum padi, and Schizaphis graminum, as well as A. gossypii in the tribe Aphidini (data not shown).] This is for the purpose of explaining cross-species amplification in the use of microsatellite loci developed from other congeneric species, A. glycines, and does not contain data. It is also not related to the research content of the subject of this study. Therefore, this sentence has been deleted with the expression "data not shown".

Reviewers' comments:

Reviewer #1: Review of PONE-D-20-16606

This study investigates the complex situation of different host races/subspecies/species within the A. gossypii group with over 500 individuals of A. gossypii and A. rhamnicola collected from 36 different plants, mainly in Korea. Mitochondrial haplotyping (COI, barcoding region) is combined with microsatellite genotyping to better understand their relationships. Of particular interest is the evolution of life-cycles, for example if host-alternating (heteroecious) taxa always give rise to monoecious taxa through the loss of the primary host, or if primary hosts can also be re-gained or changed over evolutionary time. Using Approximate Bayesian Computing to compare the likelihood of different scenarios, the authors also try to identify the ancestral primary host of the complex. The analyses confirm previous work in identifying Rhamnus as the ancestral primary host, but they indicate that counter to common belief, aphid life-cycles are quite labile. There is not only unidirectional evolution from heteroecy to monoecy. The author’s scenarios suggest that heteroecious races can derive from other heteroecious races through a shift in the primary hosts, or that primary hosts can be re-gained.

R1#1. Although I find the main results interesting, I have some concerns with this study. A lot of the results hinge on the correct assignment of plant species as primary or secondary hosts of the aphids. It is completely unclear how this was ascertained. The primary host is defined as the plant on which the aphids mate and lay their diapausing eggs. Did the authors really verify that aphids from plants identified as primary hosts were indeed reproducing sexually there, or was the timing of the sampling at least such that this could be safely assumed (late fall/early spring, just before egg laying or after hatching). It is quite common to sometimes find aphids on some woody hosts that are not necessarily their primary hosts. Misassignment of individuals to ‘their’ primary host could clearly lead to different interpretations. Please elaborate how this was excluded.

>[R1#1 Response] As the reviewer pointed out, the hosts we have identified sexuparae or fundatrix while conducting this study are Hibiscus and Rhamnus. To reflect the concerns of reviewer, we clarified the criteria for selecting plants that we have seen as primary hosts. In addition to the two obvious primary hosts, other hosts were determined to be primary hosts if the following conditions were met. 1) plants that were previously identified as the primary hosts of A. gossypii with reference to Inaizumi (1980; 1981) and Blackman and Eastop (2006), or 2) based on the lifecycle of A. gossypii on the Korean Peninsula, the time of collection is early spring (April-May) or late fall (October-November). All other hosts were set to be secondary hosts. These criteria are presented in materials and methods. In Table 1, we also inserted host type as “P: perennial, A: annual, B: biennial or annual, W: woody, H: herbaceous” in order to check our statement on the definition of primary or secondary host. (line 171)

R1#2. Completely lacking is a discussion of the relationship between the loss of sex and the loss of the primary host from the aphid life-cycle. Quite a few host-alternating aphids omit the sexual generation of their life-cycle in regions with mild climates that permit parthenogenetic overwintering or where primary host plants are not available. This will not necessarily lead to the evolution of a new host race, at least not immediately. A lot of the pest populations of A. gossypii worldwide consist of just a few permanently asexual clones with strong host associations. There were a number of studies on those by a French team led by Vanlerberghe-Masutti & colleagues, a literature that should be better integrated in this paper.

>[R1#2 Response] Thank you very much for your valuable comments you have pointed out on the core of our study. In fact, Aphis gossypii is mostly anholocylic worldwide, whereas there is host alternation in parts of E Asia and N America, with several unrelated plats utilized as primary hosts (Blackman and Eastop, 2006). Even it was found A. gossypii pops. laying eggs on some herbaceous plants (Blackman and Eastop, 2006).

Contrary to what you said, it's absolutely not that we didn't insert the contents you mentioned because we lacked understanding of "abundance of anholocyclic populations of A. gossypii in the world and loss of primary host from the host-alternating aphid". We have also been studying the speciation of the A. gossypii complex, including “evolution of gain and loss of host alternation” for many years, and in a number of previous papers including our studies, the evidences on "the relationship between the loss of the primary host from the aphid life-cycle" has been suggested in many aphid groups. There are two representative kinds of speciation mechanisms in aphids, ‘host switch between similar hosts’ like A. pisum and ‘loss of primary host (= loss of alternation)’. Maybe you mentioned the former. 

In fact, because A. gossypii has diversified in many unrelated host groups as mentioned by other great aphidologists, it could not be fully addressed by evolutionary mechanism of only host switch even though partially be explained. In particular, you may seem to be overlooking Dr. Nancy Moran's hypothesis ‘The loss of primary host’ in [19, 28, 29, 30, 35]. Although the current state of the A. gossypii aphids usually dealing with, however due to easiness of collecting, is considered to be an anholocyclic population mostly on agricultural crops, in their previous ancestors they were from the host-alternating aphid lineage, and since we think that they differentiated into anholocyclic-satellite-species by evolutionary pattern of loss of primary host, this is different from what you thought. Their permanent parthenogenesis living on the crops is a derived trait later for aphids (mentioned as an evolutionary dead end), not an ancestral state. This has been covered in many previous papers.

We have already written a discussion about what you said in the earlier draft of our manuscript, but the fundamental explanation for ‘loss of primary host’ that is out of focus on research is being redundant, so many parts have been omitted for focusing on the discussion of our new results. Deleted parts in discussion are as follows:

(If this fundamental explanation about it, we will insert these, but there are a large amount of sentences)

The evolutionary factors that had allowed aphids to live with alternating between primary and secondary host plants, as opposed to using only single host in most insects, have not yet been elucidated. Host alternation seems to be an evolutionary mechanism that is also a lifestyle strategy for overwintering of aphids, which is surprisingly closely related to their huge diversification and prosperity (Dixon, 1985a; Moran, 1992). The acquisition of the host alternation in aphidine aphids may be related to the global climate change between the Cenozoic Eocene and Miocene and the gradual but rapid fall in winter temperature (Von Dohlen and Moran, 2000; von Dohlen et al., 2006; Kim et al., 2011). In addition, the vast majorities of herbaceous plants, mainly monocotyledonous and Asteraceae plant groups, were found to their origins as Oligocene based on fossil recordings and molecular dating estimations (Kim et al., 2011). The ancestors of aphids are assumed to have settled evolutionarily on one host or host group (i.e., genus or family), which corresponds to the primary host as a primitive state (Dixon, 1985a; Stern, 1995). It is assumed that the drastic climate change and cold winter of Oligocene have resulted in a change in their living environment and the necessity of using a secondary host for various reasons (Kim et al., 2011). A close group of aphids selected for host alternation, which is highly characterized species diversity of aphids, includes polyphagous and actively speciated species representatively such as Aphis gossypii Glover (Carletto et al., 2009b) and Myzus persicae (Sulzer) (Margaritopoulos et al., 2007). Most host races in polyphagous aphid species tended to be originated from host alternating aphids, which were characterized by that the primary hosts have a strong host-parasite relationship but the small number of species are associated with aphids, while the secondary hosts are relatively very diverse attended by abundant aphids species (Mackenzie and Dixon, 1990; Blackman and Eastop, 2019). In fact, the polyphagy of aphids often appears as a complex of host races, that is, as a continuum in many plants (Carletto et al., 2009b; Peccoud et al., 2009).

Host alternation for some species often is not obligatory but facultative, in which the migration to the secondary host often can be omitted (Mackenzie, 1996; Peccoud et al., 2010). In the cases of various aphids, there are many results that host alternation appears to be a trade-off type to enhance the fitness of the group (Peccoud et al., 2010). In fact, it seems to be difficult to interpret the inherent factors of aphids that determine host alternation (Von Dohlen and Moran, 2000; Powell and Hardie, 2001). Nevertheless, through a combination of host-selectivity and trade-off in alternating between primary and secondary hosts, some groups may become isolated and genetically differentiated in the settled host, especially secondary one (Dixon, 1985a; Moran, 1992). It has been suggested that when, alternating between the primary and secondary hosts, aphids lost their primary host, they were fixed on the secondary host and then genetically isolated (Dixon, 1985a; Moran, 1992; Peccoud et al., 2010). If aphids do not return to the primary host, or if they lose genetically the return behavior of the primary, then this species will undergo a species differentiation into an incipient species as it is (Moran, 1992; Peccoud et al., 2010). Indeed, many aphid species can be seen in evolutionary dead ends through many research cases (Moran, 1988; Peccoud et al., 2010), which may be the phenomenon of loss of the primary host (Moran, 1992). Once primitive aphids originally adapted to the primary host (i.e. “ancestral” host) where they could mate and reproduce, while there may be a considerable risk in applying the same life strategy to the secondary host for adaption because the secondary host is mainly herbaceous, having a short life cycle as annual or biannual (Stern, 1995; Blackman and Eastop, 2019). Although using annual or biannual herbaceous plants is disadvantageous for aphids originally living in woody hosts, it is possible that some regulation mechanism to maintain the hibernation state such as a dwarf form (Watt and Hales, 1996) has been developed to overcome horrible condition (Lee and Kim, 2006). Fortunately, aphids using perennial herbaceous plants are capable of overwintering in the roots or bulbs of plants, thus there are aphids that lay eggs near the roots after mating , while some aphids could overwinter conveniently with dwarf forms (Lee and Kim, 2006). It is also an example that can be easily converted to anholocycly, leading to a genetic drift that accelerates an ecological isolation (Kanbe and Akimoto, 2009). Surprisingly, there are large number of anholocyclic aphids that have completely lost the sexual phase (Blackman and Eastop, 2019). If such anholocyclic aphids do not migrate to the primary host for overwinter, they will be able to continue their generation, and ultimately reach speciation (Peccoud et al., 2010). Host alternation that allows these aphids greatly adapt to the various environments is a phenomenon which is the key to understanding the evolutionary background related to diversification of aphids.

Although it is difficult to confirm whether differentiations between primary and secondary host races are due to simple host shift (Peccoud et al., 2009) or the loss of primary host (Moran, 1992) because there was no previous study to substantially perform genetic comparison between hetoroecious holocyclic host (primary host) races and anholocyclic host (secondary host) races, our results support the possibility of loss of the primary host more in the gossypii group. The isolation effect of anholocycly leading to speciation has already been studied in many species of the aphidine genera such as Acyrthosiphon, Myzus and Cryptaphis as well as Aphis (Dixon, 1987; Carletto et al., 2009b; Kanbe and Akimoto, 2009; Peccoud et al., 2010). Possibly in the host alternation, if isolated entities do not migrate anymore to the overwintering host and then are isolated in secondary host, genetic isolation by genetic drift will appear much faster, which will allow species differentiation to progress more quickly. If some host race or population is isolated and still holocyclic, i.e., having sexual phase, in the secondary host, it is a big stumbling block for surviving that it should omit the process of finding the overwintering host for reproduction. Without host alternation, it is doubtful that it can be reproduced in the secondary host whether in facultative or obligatory anholocycly. Therefore, it is important to know how the holocycly is possible in the herbaceous plant, or whether they should return to the overwintering host eventually after dramatically surviving the anholocyclic state over the course of a year or several years. … - omitted -

Of course, we highly value the work of our esteemed Vanlerberghe-Masutti & colleagues, and impressed by their work, thus, we conducted this study to explore the population genetic novelty and taxonomic problems of the A. gossypii–rhanmicola complex. We read most of Vanlerberghe-Masutti & colleagues' research papers and conducted research based on our understanding of those contents. As a result, in this study, we were able to cite their valuable studies through references [11, 14, 36, 42, and 50] and get ideas and hypotheses about the population genetics work. In the main text, the related contents are included in the introduction and discussion. Unlike the most recent study by Carltto et al. (2009) with the samples only from asexual lineages, our study contains most of the wildtype HAPs inhabiting wild hosts including host-alternating populations, so it is differentiated from previous studies and can be said to be an extended study.

With regard to your opinion of “A lot of the pest populations of A. gossypii worldwide consist of just a few permanently asexual clones with strong host associations.”, you're definitely right, but there are still a number of A. gossypii HAPs in wildlife that we don't know about, and that's being covered and studied in the samples in this paper. We still don't know how many unsampled and unrecognized wild HAPs exist in this A. gossypii–rhanmicola complex. Several host lines found in crop hosts form the dominance of the A. gossypii population, but that is a very small fraction of the genetic diversity. Our paper broadly covers the existence of these wild HAPs and the linkages between them and their sexual and asexual lineages, furthermore, the potential for differentiation with HAPs of A. rhamnicola.

Since there is a study that has already been verified in the paper of Vanlerberghe-Masutti & colleagues who have already spoken about this, the related contents have already been cited from those references [11, 14, 36, 42, and 50]. 

R1#3. My main concern, however, is the presentation of the population genetic analyses and partially their (over-) interpretation. The Results section is extremely hard to follow and requires major changes to make it more accessible to an average reader. Some English language editing would also help with that.

>[R1#3 Response] Since the authors are not English native, I admit that there is a little awkward expression in the description, but it has been confirmed that there is no big problem in delivering the results. In addition,

This manuscript was proofread and edited by the professional English editors of Scientific English Research Paper Editing Service at HARRISCO, Company (Certificate no. E_200428_02). We have already attached the certificate pdf as supporting information.

R1#4. My issues start with the standard population genetic analyses of the microsatellite data and their interpretation. First of all, everything on lines 433-441 is completely speculative without supporting evidence. There is simply no way of telling just from the patterns whether significant deviations from HWE in some subpopulations are due heterosis or any other mechanism. Some other statements are plain wrong, e.g. that “an increase in heterozygosity that was generally due to random mating or outbreeding”. Random mating is what restores HWE in a population! I think the authors would better restrict themselves to the description of the patterns. Secondly, some deviations from HWE may simply be due to the inclusion of multiple copies of the same genotype (clone). Generally, clonal diversity is high in these samples, but in Ag_CA, for example, there are just six different MLGs among 25 individuals. This sample cannot be in HWE for purely statistical reasons. If just one relatively heterozygous genotype occurs multiple times in this sample, there is likely to be a significant heterozygote excess. I would thus recommend to test for deviations from HWE also with a dataset reducing clonal copies, i.e. with only one representative of each MLG per sample.

>[R1#4 Response] We fully agree with your comment. We tested again using a reduced data set containing only one copy of each MLG in HWE, and insert like this “Because the clonal copies of MLGs due to the parthenogenetic life cycle of aphids could affect and distort the estimation of HWE (Sunnucks et al. 1997), we used a reduced data set containing only one copy of each MLG when estimating HWE.” And “Several assumptions of HWE can still be violated, thereby these estimates are used only for descriptive purposes even although the clonal MLG copies were removed from data analysis (Sunnucks et al. 1997).“ Nevertheless, even after reanalysis excluding the MLG clonal copies, our results interestingly remained unchanged in HWE. Thank you. (line 281)

R1#5. Then I find the verbal account of the pairwise genetic differentiation (Fst) results very hard to follow (l. 444-457). The second sentence, for example, makes no sense to me. Why pick out four particular populations for “the HAPs of A. gossypii”, calculate some average Fst between them and A. rhamnicola (which populations, all of them?), and then only come up with three values? I really cannot follow. The whole business of somehow averaging pairwise Fst values is very confusing. Please re-structure this whole passage. Maybe you can get by describing the main patterns from Table 2 rather than work with some difficult to trace averages.

>[R1#5 Response] In population genetics studies, providing the Fst value between each population is the most basic result. Therefore, the contents of Table 2 should be presented as appropriate, although it is difficult to identify pairwise between groups. In addition, in order to confirm the comparison of these pairwise Fst values for each of the species group HAPs, we provided averaging pairwise Fst values in the text. Therefore, averaging pairwise Fst provided as summarized into the three groups: all HAPs, HAPs only in A. gossypii, and HAPs in A. rhamnicola, for the purpose of making it possible to know the genetic differences between HAPs from the values of the two species. In addition, since the degree of differentiation due to their genetic variation can be inferred compared to the results of AMOVA, averaging pairwise Fst is not considered unnecessary information. The next presented results are averaging pairwise Fst for each host plant genus or family group, and comparison results showing how genetically close or distant the host associated population groups defined by each genus or familiy are. These are the descriptive theorems necessary to understand the degree of differentiation between HAPs according to host specificity within A. gossypii and A. rhamnicola, respectively. Some descriptions have been revised and the opinions pointed out by reviewers have been reflected. Thank you. (line 468)

R1#6. Similarly confusing is the passage reporting the AMOVA results (l. 458 – 465). The groupings are not properly explained, neither in the Methods nor here. The number of df in the AMOVA table suggests there were four groups for ‘host plant’, but what were these? The same four picked in the paragraph above (for unknown reasons), or was it plant genera/families (Cucurbitaceae, Solanaceae, Euonymus, Asteraceae)? Please clarify.

>[R1#6 Response] I’m very sorry. As you mentioned, groupings were not appropriated for host plant and geographic isolation in this study. Therefore, we performed again AMOVA for microsatellite data analysis of aphids grouped by three cases: (1) gossypii vs rhamnicola, (2) perennial vs non-perennial host groups in A. gossypii, (C) perennial vs non-perennial host groups in A. rhamnicola, resulting in Table 3. The AMOVA results are changed like this: In the case of the analysis grouped by case 1, percentages of the genetic variance (PV) ‘among groups’ and ‘among populations within groups’ were 14.59 % and 22.60 %, respectively, which shows that there is some grouping effect by host plants, even though the majority of genetic variation was found ‘among individuals within populations’ as approximately 63 %. However, the genetic variance of about -1 ~ 0 % ‘among groups’ in the both analyses grouped by cases 2 and 3 suggests that there are no grouped structures according to their lives in the perennial or non-perennial hosts on both A. gossypii and A. rhamnicola. Interestingly, PV of ‘among populations within groups’ in A. rhamnicola was about 20 % higher than that in A. gossypii, which means that the HAPs of A. rhamnicola is genetically differentiated further than those of A. gossypii. (line 481) (line 501-table 3)

R1#7. In the passage reporting the results of the assignment tests with GENECLASS, it is unclear what the first values in the brackets before the self-assignment probabilities (SA) represent and why they are relevant. Please explain.

>[R1#7 Response] GENECLASS 2 were carried out to identify the host-associated population (HAP) membership of 578 individuals from all the 36 HAPs. In the sentence “In A. gossypii, the mean assignment probability from 391 A. gossypii individuals into Ag_RH had the highest value (0.446, SA = 0.381)”, the first value in the parenthesis means the ‘mean assignment probability from 391 A. gossypii individuals into HAP of Ag_RH’, which was calculated by average of individual assignment values when assigning A. gossypii to Ag_RH. It was hypothesized that not-self-assignment value should be strongly relevant to the primary (even lost) or ancestral host, thus the highest value would be observed if some host was the most likely ancestral host of all A. gossypii HAPS. Therefore, in the comparison, assigning into Ag_RH had the highest value (0.446, SA = 0.381), which was followed by the assignment value into each reference HAP of Ag_HI (0.219, SA = 0.478) and Ag_PU (0.214, SA = 0.458) from all A. gossypii individuals, and SA also indicated to compare both mean values (i.e. assignment to other or to self). (line 550)

R1#8. Finally, the results text on the ABC analysis comparing different evolutionary scenarios is very hard to follow. There are literally two full pages of sentences like these: “Scenario A3 showed a PP ranging (0.313 (nδ = 8 000) to 0.290 (nδ = 80 000)), with a 95 % CI of (0.251–0.375) and (0.270–0.309). Scenario A4 showed a PP ranging (0.001 (nδ = 8 000) to 0.001 (nδ = 80 000)), with a 95 % CI of (0.000–0.002) and (0.001–0.001).” With the corresponding figures all hidden in the electronic appendix, this is all but unreadable. The results are interesting, so my suggestion would be to maybe present the analysis results in the form of a table, and combine this with a figure at least of the best-supported evolutionary scenario in the paper, not the appendix (like the different plots in Fig S1). This would make the results more accessible. A clearer explanation of the nδ numbers is also required (number of simulated datasets considered). Why look at two different numbers, and why do these numbers not correspond to those mentioned in the Methods section?

>[R1#8 Response] 

I'm very sorry. I agree to the poor readability of the ABC results. All of these contents are organized in the Tables 3, which is inserted in the main text, thereby all the relevant contents (unreadable sentences) that present values and numbers in the text have been deleted. In addition, ‘nδ’ means the number from the selected simulation datasets, which matched the description in M&M. However, ‘nδ’ was already defined in M&M (line 309) as “DIYABC generates a simulated data set that is then used to select those most similar to the observed data set, and the so-called selected data set (nδ), which are finally used to estimate the posterior distribution of parameters”. In most studies, suggesting the results of the logistic regression analyzed by ABC, two different numbers were indicated, which means both the 1% selected simulated datasets of the initial (nδ = 8000 or 6000) and final (nδ = 80 000 or 60 000) simulation datasets representative of all the simulated datasets. All the correction has been completed as you pointed out in M&M and Results. (Line 337)

R1#9. Issues of over-interpretation also extend to the discussion, for example “they were clearly identified as disctinct species, based on the microsatellite analysis”. There is not really an established straightforward way inferring species status just from genetic differentiation at microsatellite loci.

>[R1#9 Response] I admit that there are some problems with the expression describing the results. Corrected this expression as follows: "Comparing the host races between A. gossypii and A. rhamnicola, most of them were distributed in two haplotypes (H9 and H2), although their HAPs were clearly separated as a distinct species-group based on the PCoA analysis using microsatellite loci (Fig 2). "Since the HAPs of A. gossypii and A. rhamnicola are accurately separated in PCoA, it was confirmed that there is no abnormality in the content. In M&M, the related sentences are included like this: “However, there are a lot of the haplotypes cross-shared between A. gossypii and A. rhamnicola (See Results). In this case, exceptionally we applied the dominant assignment (white or green) of the genetic structure (K =2) by STRUCTURE and the PCoA results for the species identification (see Results).” (Line 213)

R1#10. Although the authors declare that all data will be publicly available, I could not find any statement in the PDF about where the data are or will be made accessible.

>[R1#10 Response] In Plos One's submission, there was no function to attach public accessible data like DRYAD, so it was not disclosed. There was an attachment mistake in submitting the input data file. I apologize for this. Input files for public use are organized as follows, and added as supporting information in submission:

★Aphis spp. 578 individuals STRUCTURE INPUT FILE.txt

★Aphis spp. 578 individuals ARLEQUIN INPUT FILEs.arp

★Aphis spp. 578 individuals DISTRUCT PARAMETER FILE.zip

★Aphis spp. 578 individuals GENALEX INPUT FILE.xlsx

★Aphis spp. 578 individuals GENEPOP GENECLASS2 INPUT FILE.txt

R1#11. l. 26: tested to confirm -> used to infer

>[R1#11 Response] corrected

R1#12. l. 27: most primitive -> ancestral

>[R1#12 Response] corrected. All ‘primitive’ words are changed to ‘ancestral’. Thanks.

R1#13. l. 28: delete ‘respectively’

>[R1#13 Response] deleted

R1#14. l. 31 (and elsewhere): heteroecy (noun) -> heteroecious (adjective)

>[R1#14 Response] corrected

R1#15. l. 34: delete ‘of counterpart species’

>[R1#15 Response] deleted

R1#16. l. 48: Jaenike 1990 in AnnuRevEcolSyst also seems like a key reference here.

>[R1#16 Response] inserted. Thank you.

R1#17. l.85-87: Unclear sentence. Please re-word.

>[R1#17 Response] These are re-worded as “Approximately 10 % of 5,000 aphid species exhibit the seasonal host alternation (i.e. heteroecy) between primary and secondary hosts, which mysteriously are comprised with a set of phylogenetically unrelated host plants [22, 27, 28]. In addition, among all phytophagous insects, the complex life cycle completed by multiple generations is known to be limited to the aphids (Aphidoidea) [29, 30].”

R1#18. l. 110: what does ‘primitive’ mean in this context?

>[R1#18 Response] corrected. ‘primitive’ is changed to ‘ancestral’. Same to [R1#12 Response]

R1#19. l. 111 and elsewhere: adaptive -> adapted

>[R1#19 Response] ‘adaptive’ is suitable because of being contrast to ‘maladaptive’. ‘adapted’ means ‘already live in the host’ but ‘adaptive’ means ‘applicable to use as a (primary) host’ in this sentence.

R1#20. l. 131: Again, I think ancestral would be more appropriate than primitive.

>[R1#20 Response] corrected. Thank you.

R1#21. Table 2: Host-race populations may be an undue inference. Maybe just call them ‘host-associated’?

>[R1#21 Response] corrected. Thank you.

R1#22. Fig. 3: If color-coding points with reference to the STRUCTURE plot for K=3 in Fig. 4, why not use the 

same colors for all groups?

>[R1#22 Response] According to your advice, plot group color-code in PCoA of Fig. 3 and assignment group color-code in STRUCTURE (K=3) of Fig. 4 has been modified to match each other. Thank you.

R1#23. l. 514-515: One of these K should be something other then 5, right?

>[R1#23 Response] Oh, it was a typo. It is corrected as “When K = 5, the genetic structure was basically similar to that at K = 4...” Thank you.

R1#24. l. 547: likelihood -> likely

>[R1#24 Response] corrected. Thank you.

R1#25. l. 720-721: Just like this the statement is incorrect (does not often migrate…). Aphis fabae is also a complex of subspecies with a shared primary host and different secondary host ranges, but the vast majority of them does migrate routinely between primary and secondary host, at least in climates with a cold winter.

>[R1#25 Response] This sentence is corrected as “As an example, a facultative alternation lifecycle has been reported in populations of Aphis fabae, even although the vast majoriy of them migrate routinely between primary and secondary hosts.” Thank you.

Reviewer #2: This study focusses on two species of aphids, A. gossypii and A. rhamnicola. Both are members of a confusing species complex, the frangulae group, many of which use Rhamnus as a primary host. The authors newly sampled many populations of both species from a much broader range of host plants than has been done before. They conducted various population genetic analyses from mitochondrial COI barcode sequences and from multiple microsatellite loci. One goal was to determine whether populations might fall into discrete genetic entities that use specific host plants (or specific sets of host plants). Another goal was to identify the pattern of shifts between host plants and implications for life cycle evolution, e.g., whether heteroecious life cycles could be derived directly from other heteroecious life cycles on a different primary host. A final goal was to infer the ancestral primary host plant.

R2#1. I think a major contribution of this work is in the identification of the host-associated populations—that is, that both of these species sort out into biotypes that are fairly specific to certain host plants. They are not each a randomly, highly polyphagous entity. Except for the COI haplotype network and principal coordinate methods, I was unfamiliar with the data analysis for microsatellites, so I can’t comment on the specifics of those, other than one point (below). The haplotype and PCoA methods seemed fine.

>[R2#1 Response] We used standard methodology widely used in previous research papers in the field of population genetics using microsatellite loci. PCoA, Structure, GenClass2 assignment and DIYABC are the previously proven methods in such group genetics research. Thank you.

R2#2. One point in the ABC methods and analysis that concerns me is setting A. rhamnicola in the ancestral position of the genealogy (lines 323-325). This was based on previous findings of reference #41 (Lee Y, Lee W, Lee S, Kim H. A cryptic species of Aphis gossypii (Hemiptera: Aphididae) complex revealed by genetic divergence and different host plant association. Bull Entomol Res. 2015;105(1):40-51). However, the tree in that paper is an unrooted neighbor-joining distance dendrogram, not a true character-based rooted phylogeny. That aside, what is more pertinent is that A. rhamnicola is not located in a basal position in that tree, but is nested in a more derived position—but actually since the tree is technically unrooted we don’t really know where A. rhamnicola is placed relative to the root. Furthermore, nearly all of the inter-species relationships in the tree are unsupported, with bootstrap values below 60-50%. If the ABC results are dependent on which entities are designated as “ancestral” then the findings from these tests could be in question. I urge the authors to acknowledge the uncertainty in relationships in this species group and determine if it affects their results.

>[R2#2 Response] For information on the ancestral position of A. rhamnicola, please see reference #40 (Kim H, Lee S, Jang Y. Macroevolutionary patterns in the Aphidini aphids (Hemiptera: Aphididae): diversification, host association, and biogeographic origins.PLoS One.2011;6(9):e24749.) as well as #39 (Kim H, Hoelmer KA, Lee W, Kwon YD, Lee S. Molecular and morphological identification of the soybean aphid and other Aphis species on the primary host Rhamnus davurica in Asia. Annals of the Entomological Society of America. 2010;103(4):532-43.), not reference #42 (Lee et al. 2015). However, in the ref #40 paper, since the sample of A. rhamnicola was studied before being described as a new species, It was specified as A. sp. ex Rhamnus sp.1 in the phylogenetic tree in ref #40. In ref #40, with a high support value, A. rhamnicola is located in the basal position, and A. gossypii appears in the nested position with other sister species (A. sumire, A. clerodendri, etc.). According to molecular dating analysis, the gosypii group relative clade was nested inside the clade of the common ancestor with A. rhamnicola, and more recently, gossypii and its sister species diverged. As mentioned, ref #42 shows such a relationship unrooted-like because A. sumire and A. clerodendri, which form a sister relationship with A. gossypii, were excluded in the phylogeny. As a revision for this, I'll cite that with ref #39, #40 instead of #42 (removed from that citation). Thank you.

R2#3. In the Discussion, the authors interpret the cross-sharing of haplotypes and microsat alleles as hybridization. However, given that the relationships between gossypii, rhamnicola, and related species are uncertain (see above), it seems that another explanation for shared haplotypes and alleles might be incomplete lineage sorting. The authors should consider this (and discount if they can).

>[R2#3 Response] We have studied in a number of papers since our previous work that there are few genetic differences between A. gossypii and its closely related species (A. sumire, A. clerodendri, A. sedi. A. egomae, etc.), and that lineage sorting is very difficult as you know. The reason why A. gossypii has so many synonyms historically can be seen through this paper. In this study, it was newly found that in COI haplotype, two related species, A. gossypii and A. rhamnicola, cross-share each other's haplotypes. In fact, we also found cross-sharing of COI haplotype (mtDNA haplotype) between A. glycines and A. rhamnicola. This phenomenon is detected for the first time in aphid group. Although we performed COI analysis initially to limit their species, as the result, the species were not separated on the haplotype, that is the species were not separated on the COI haplotype, but rather shared and mixed with each other. Limiting the boundary of the species with the COI haplotype It was impossible. At first, there was a suspicion of misidentification of the two species with no morphological difference, but it was confirmed that the species boundary appeared correctly when microsatellite was used. Based on using microsatellite loci, A. gossypii samples were clearly separated from those of A. rhamnicola in the results of PCoA, Structure, Genclass2 assignment and etc. Therefore, we determined the species based on the results of microsatellite analysis, applying the species limitation of the two species in this paper. 

R2#4. • avoid the term “primitive”. Use “ancestral” instead.

>[R2#4 Response] corrected. Same to [R1#18 Response]

R2#5. • Figure 2B caption needs correcting (it repeats 2A caption)

>[R2#5 Response] YL

R2#6. • Is the microsatellite raw data deposit somewhere?

>[R2#6 Response] Yes it is deposited in supporting information. Same to [R1#10 Response]

---

## [Decision Letter · Decision Letter 1]

30 Nov 2020

PONE-D-20-16606R1

Complex evolution in Aphis gossypii group (Hemiptera: Aphididae), evidence of primary host shift and hybridization between sympatric species

PLOS ONE

Dear Dr. Kim,

Thank you for submitting your manuscript to PLOS ONE. After careful consideration, we feel that it has merit but does not fully meet PLOS ONE’s publication criteria as it currently stands. Therefore, we invite you to submit a revised version of the manuscript that addresses the points raised during the review process.

Reviewer 1 continues to be concerned with over- or mis-interpretation of your population genetics data.  You should address the following points in your revision:

The conclusions you draw about the cause of heterozygote excess or heterozygote deficit (lines 455-464) are not appropriate for the Results section.  If retained, this section should be moved to the Discussion, and the comments of Reviewer 1 addressed.The potential effects of incomplete lineage sorting should not be ignored in the Discussion.  You should consider stating that incomplete lineage sorting cannot be discounted as a cause before revealing that interspecific matings have been observed (which gives more credence to the hybridization hypothesis).

You might also consider:

Providing a citation and/or example species in support of the statement in lines 632-634.

We look forward to receiving your revised manuscript.

Kind regards,

Owain Rhys Edwards, Ph.D.

Academic Editor

PLOS ONE

Reviewers' comments:

Reviewer's Responses to Questions

**Comments to the Author**

1. If the authors have adequately addressed your comments raised in a previous round of review and you feel that this manuscript is now acceptable for publication, you may indicate that here to bypass the “Comments to the Author” section, enter your conflict of interest statement in the “Confidential to Editor” section, and submit your "Accept" recommendation.

Reviewer #1: (No Response)

Reviewer #2: All comments have been addressed

2. Is the manuscript technically sound, and do the data support the conclusions?

Reviewer #1: Partly

Reviewer #2: Yes

3. Has the statistical analysis been performed appropriately and rigorously? 

Reviewer #1: I Don't Know

Reviewer #2: Yes

4. Have the authors made all data underlying the findings in their manuscript fully available?

Reviewer #1: Yes

Reviewer #2: Yes

5. Is the manuscript presented in an intelligible fashion and written in standard English?

Reviewer #1: No

Reviewer #2: Yes

6. Review Comments to the Author

Reviewer #1: Summarizing the ABC analysis in a table made this paper much more readable, and the new explanation of how plants were defined as either primary or secondary hosts is helpful. Consider indicating in Table 1 which plants were considered to be primary or secondary hosts in the analyses (now only those already described as primary hosts in the literature are marked as such).

In other respects, this revision failed to address justified criticism by the reviewers. Reviewer 2 made the perfectly valid point that shared mitochondrial haplotypes between A. gossypii and A. rhamnicola (and between other potential species showing strong differentiation with nuclear markers) could be the result of incomplete lineage sorting rather than evidence of ongoing hybridization. I cannot judge which explanation is more likely, and the authors have every right to discuss why they consider hybridization more likely based on the available evidence. But this is not done in the paper. There is some unconvincing rebuttal in the cover letter and the term incomplete lineage sorting does not even show up in the paper’s Discussion. This is not thorough.

Similarly, reviewer 1 criticized the misleading interpretation of some standard population genetic indices (heterozygosities etc.). While the authors did check whether some of the deviations from HWE might be due to the inclusion of multiple clonal copies of the same genotypes within populations, the (over-)interpreted summary of the results remained completely unchanged in the paper. For example this part:

“Heterozygote excess in Ag_CA, Ag_CP, and Ar_PE were likely the result of heterosis or over-dominance related to selection preference toward heterozygous combination [81], or fixation of heterozygous genotypes; and, correspondingly, negative FIS values also showed an increase in heterozygosity that was generally due to random mating or outbreeding [82]. In contrast, heterozygote deficit (i.e., homozygote excess) in Ag_CJ, Ar_YO, Ar_PH, and Ar_RU was likely caused by retaining numerous unique genotypes with private alleles within a population, and positive FIS values explained that the amount of heterozygous offspring in the population decreased, usually due to inbreeding [82].”

This is scientifically unsound in several respects. Without other, independent evidence it is simply not possible to infer heterosis as the cause of heterozygote excess at neutral markers. Random mating will restore HWE and not lead to “an increase in heterozygosity”. Etc.

Reviewer #2: I have no further comments concerning this manuscript. THe authors have addressed my concerns in this revision.

7. PLOS authors have the option to publish the peer review history of their article (what does this mean?). If published, this will include your full peer review and any attached files.

Reviewer #1: No

Reviewer #2: No

---

## [Author Response · Author response to Decision Letter 1]

1 Jan 2021

[1]

Reviewer 1: The potential effects of incomplete lineage sorting should not be ignored in the Discussion. You should consider stating that incomplete lineage sorting cannot be discounted as a cause before revealing that interspecific matings have been observed (which gives more credence to the hybridization hypothesis).

Editor: In other respects, this revision failed to address justified criticism by the reviewers. Reviewer 2 made the perfectly valid point that shared mitochondrial haplotypes between A. gossypii and A. rhamnicola (and between other potential species showing strong differentiation with nuclear markers) could be the result of incomplete lineage sorting rather than evidence of ongoing hybridization. I cannot judge which explanation is more likely, and the authors have every right to discuss why they consider hybridization more likely based on the available evidence. But this is not done in the paper. There is some unconvincing rebuttal in the cover letter and the term incomplete lineage sorting does not even show up in the paper’s Discussion. This is not thorough.

>>

[Response and rebuttal]

We have reflected in the manuscript what the reviewer and editor pointed out. As pointed out by reviews comments, doing lineage sorting by COI haplotype results is likely unclear at the first submitted manuscript. Since A. gossypii and A. rhamnicola cross-sharing haplotypes in several host plants, there is a problem with lineage sorting or DNA identification with only COI haplotypes (barcoding). However, I am not saying clearly that our sample is misidentified with the two species, A. gossypii and A. rhamnicola. In particular, it was revealed that genotyping by microsatellite was the only way to distinguish between these two evolutionarily intermediate species, even for organizing the sampled individuals in HAP type and utilizing them for analysis.

Accepting these editor and reviewer’s opinions, the identification content by COI haplotype (barcoding) was deleted and supplemented with the concept of species lineage (group) sorting instead. Therefore, ‘Groups’ sorted by lineage in the table are specified and subdivided into A. gossypii Group1, A. g. Group 2, A. rhamnicola Group 1, A. r. Group 2, A. r. Group 3. In addition, according to our new lineage sorting, species names were deleted in Fig 2 and Fig 4. since specifying the species creates a rather confusing situation and misunderstanding of original purpose of study. The changes in the text are as follows.

"The two Aphis species, A. gossypii and A. rhamnicola, we study here are not only very similar in morphology, but also share several host plants due to the polyphagy. Although we performed species identification through morphology and host plant relationships as a first step and also tested DNA barcoding for all individuals collected on their shared host plants (eg Capsella, Rhamnus, and Rubia), we found that there were a lot of the haplotypes cross-shared between A. gossypii and A. rhamnicola (see Results). Therefore, instead of identifying the species with 36 HAPs, we applied the dominant assignment (white, green, blue, red, dark blue) of the genetic structure (K =3, 4, 5) by STRUCTURE as well as the PCoA results (see Results) to sort their lineags into five groups as Aphis gossypii Group 1, A. g. Group 2, A. rhmanicola Group 1, A. r. Group 2 and A. r. Group 3 (Table 1). 'Aphis gossypii' and'A. rhamincola', which are mentioned later, are meant to include all group lin eages containing the HAPs assigned by the results. Table S3 shows detailed information for lineage sorted samples used in DNA analyses." -> LINE 202-211, Table 1, Fig2, Fig4 revised

In addition, the content of HAP mixing in the intermediating host plant based on COI or 16S hybridization is sufficiently mentioned in the discussion by citing a paper by Lozier et al. [89]. Please check the contents at the end of the discussion. In relation to COI hybridization, hybridized population of Hyalopterus pruni has already found by Lozier et al. [89]. It has been explored and studied by and presented the second observation in aphids and corroborating results in our study. It is emphasized once again that the phenomenon of cross-sharing of COI haplotypes between the two species is by no means due to errors in lineage sorting.

 

[2]

Reviewer 1 continues to be concerned with over- or mis-interpretation of your population genetics data. You should address the following points in your revision:

The conclusions you draw about the cause of heterozygote excess or heterozygote deficit (lines 455-464) are not appropriate for the Results section. If retained, this section should be moved to the Discussion, and the comments of Reviewer 1 addressed.

Editor: Similarly, reviewer 1 criticized the misleading interpretation of some standard population genetic indices (heterozygosities etc.). While the authors did check whether some of the deviations from HWE might be due to the inclusion of multiple clonal copies of the same genotypes within populations, the (over-)interpreted summary of the results remained completely unchanged in the paper. For example, this part: 

“Heterozygote excess in Ag_CA, Ag_CP, and Ar_PE were likely the result of heterosis or over-dominance related to selection preference toward heterozygous combination [81], or fixation of heterozygous genotypes; and, correspondingly, negative FIS values also showed an increase in heterozygosity that was generally due to random mating or outbreeding [82]. In contrast, heterozygote deficit (i.e., homozygote excess) in Ag_CJ, Ar_YO, Ar_PH, and Ar_RU was likely caused by retaining numerous unique genotypes with private alleles within a population, and positive FIS values explained that the amount of heterozygous offspring in the population decreased, usually due to inbreeding [82].”

This is scientifically unsound in several respects. Without other, independent evidence it is simply not possible to infer heterosis as the cause of heterozygote excess at neutral markers. 

Random mating will restore HWE and not lead to “an increase in heterozygosity”. Etc.

>>

[Response and rebuttal]

Because aphids ‘parthenogenetically’ reproduce in their secondary host, especially under anholocyclic life, the result of heterosis or over-dominance related to selection preference toward heterozygous combination or fixation of heterozygous genotypes have been commonly reported in study of aphids. This is far from the normal reproductive situation under random mating on other diploid organisms. We cited additional references. Please see citations below:

81. Delmotte F, Sabater-Munoz B, Prunier-Leterme N, Latorre A, Sunnucks P, Rispe C, et al. Phylogenetic evidence for hybrid origins of asexual lineages in an aphid species. Evolution. 2003;57(6):1291-303. doi: Doi 10.1554/02-557. PubMed PMID: WOS:000183997400007.

82. Simon JC, Baumann S, Sunnucks P, Hebert PDN, Pierre JS, Le Gallic JF, et al. Reproductive mode and population genetic structure of the cereal aphid Sitobion avenae studied using phenotypic and microsatellite markers. Molecular Ecology. 1999;8(4):531-45. doi: DOI 10.1046/j.1365-294x.1999.00583.x. PubMed PMID: WOS:000080177700003.

83. Delmotte F, Leterme N, Gauthier JP, Rispe C, Simon JC. Genetic architecture of sexual and asexual populations of the aphid Rhopalosiphum padi based on allozyme and microsatellite markers. Molecular Ecology. 2002;11(4):711-23. doi: DOI 10.1046/j.1365-294X.2002.01478.x. PubMed PMID: WOS:000175250300007.

84. Wilson ACC, Sunnucks P, Blackman RL, Hales DF. Microsatellite variation in cyclically parthenogenetic populations of Myzus persicae in south-eastern Australia. Heredity. 2002;88(4):258-66. doi: 10.1038/sj.hdy.6800037.

In the respects to the description of heterozygote deficit, in the contrary to parthenogenetic life, sexual lineages generally show the heterozygote deficits [81-84]. However, as you pointed out, I cannot connect it without confirming clear evidence of correlation between sexual lineages and ecology of HAPs sampled from our study, thus deleted it (explanation of heterozygote deficits. Finally, the sentences were amended as below:

“Heterozygote excess in Ag_CA, Ag_CP, and Ar_PE were likely the result of heterosis or over-dominance related to selection preference toward heterozygous combination or fixation of heterozygous genotypes due to parthenogenesis of aphids in secondary host, especially under anholocyclic (permanently asexual) life [81]. Similar to our results, this phenomenon was already reported from several aphid species such as Sitobion avenae, Myzus persicae and Rhopalosiphum padi having permanently or temporary asexual life, which showed the significant heterozygote excess [82-84]. Negative FIS values also showed an increase in heterozygosity that was generally due to random mating or outbreeding, whereas positive FIS values explained that the amount of heterozygous offspring in the population decreased, usually due to inbreeding [85].” -> LINE 459-466 revised

Unlike the reviewer's request, because we do not treat discussion of the heterozygote excess in Ag_CA, Ag_CP, and Ar_PE, we still remain these contents in the results section in view of consistency of manuscript content.

[3]

You might also consider:

Providing a citation and/or example species in support of the statement in lines 632-634.

Add the related references to support the statement:

“This is similar to the tendency found in most polyphagous aphids that the primary host is shared, but the secondary host is completely different [32, 86-89].”

32. Moran NA. The Evolution of Aphid Life-Cycles. Annual Review of Entomology. 1992;37:321-48. doi: DOI 10.1146/annurev.ento.37.1.321. PubMed PMID: WOS:A1992GY50200014.

86. Lozier JD, Roderick GK, Mills NJ. Genetic evidence from mitochondrial, nuclear, and endosymbiont markers for the evolution of host plant associated species in the aphid genus Hyalopterus (Hemiptera : Aphididae). Evolution. 2007;61(6):1353-67. doi: DOI 10.1111/j.1558-5646.2007.00110.x. PubMed PMID: WOS:000247125100009.

87. Margaritopoulos JT, Tsitsipis JA, Goudoudaki S, Blackman RL. Life cycle variation of Myzus persicae (Hemiptera: Aphididae) in Greece. Bulletin of Entomological Research. 2002;92(4):309-19. Epub 2002/08/23. doi: 10.1079/BER2002167

S0007485302000366 [pii]. PubMed PMID: 12191439.

88. Guldemond JA. Host Plant Relationships and Life-Cycles of the Aphid Genus Cryptomyzus. Entomologia Experimentalis Et Applicata. 1991;58(1):21-30. PubMed PMID: WOS:A1991EZ54000003.

89. Jousselin E, Genson G, Coeur d'Acier A. Evolutionary lability of a complex life cycle in the aphid genus Brachycaudus. BMC evolutionary biology. 2010;10(1):295. doi: 10.1186/1471-2148-10-295. PubMed PMID: 20920188; PubMed Central PMCID: PMCPMC2958166.

---

## [Editor Report · Decision Letter 2]

5 Jan 2021

Complex evolution in Aphis gossypii group (Hemiptera: Aphididae), evidence of primary host shift and hybridization between sympatric species

PONE-D-20-16606R2

Dear Dr. Kim,

We’re pleased to inform you that your manuscript has been judged scientifically suitable for publication and will be formally accepted for publication once it meets all outstanding technical requirements.

Kind regards,

Owain Rhys Edwards, Ph.D.

Academic Editor

PLOS ONE

Additional Editor Comments (optional):

Focusing on the sorting of the "Group" lineages without reference to the species identifications was a very good idea, as the issue of incomplete lineage sorting no longer appears.

You have also dealt sufficiently with the issue of heterozygote excess.

I have attached a version with some comments, all of which relate to improving on the English to improve clarity.
---

## [Editor Report · Acceptance letter]

13 Jan 2021

PONE-D-20-16606R2 

**Complex evolution in *Aphis gossypii* group (Hemiptera: Aphididae), evidence of primary host shift and hybridization between sympatric species**

Dear Dr. Kim:

I'm pleased to inform you that your manuscript has been deemed suitable for publication in PLOS ONE. Congratulations! Your manuscript is now with our production department. 

Kind regards, 

on behalf of

Dr. Owain Rhys Edwards 

Academic Editor

PLOS ONE